# Sea Ice Assimilation into a Coupled Ocean-Sea Ice Model Using its Adjoint

Nikolay V. Koldunov[1,2,3], Armin Köhl[1], Nuno Serra[1], and Detlef Stammer[1]

[1]Institut für Meereskunde, Centrum für Erdsystemforschung und Nachhaltigkeit, Universität Hamburg, Germany
[2]MARUM - Center for Marine Environmental Sciences, Bremen
[3]AWI - Alfred Wegener Institute for Polar and Marine Research, Bremerhaven, Germany

*Correspondence to:* Nikolay Koldunov (nikolay.koldunov@awi.de)

**Abstract.** Satellite sea ice concentrations (SIC), together with several ocean parameters, are assimilated into a regional Arctic coupled ocean-sea ice model covering the period 2000-2008 using the adjoint method. There is substantial improvement in the representation of the SIC spatial distribution, in particular with respect to the position of the ice edge and to the concentrations in the central parts of the Arctic Ocean during summer months. Seasonal cycles of total Arctic sea ice area show an overall improvement. During summer months, values of sea ice extent (SIE) integrated over the model domain become underestimated compared to observations, however absolute differences of mean SIE to the data are reduced in nearly all months and years. Along with the SIC, the sea ice thickness fields also become closer to observations, providing added-value by the assimilation. Very sparse ocean data in the Arctic, corresponding to a very small contribution to the cost function, prevent sizable improvements of assimilated ocean variables, with the exception of the sea surface temperature.

## 1 Introduction

The Arctic region is expected to experience a dramatic anthropogenic temperature increase over the years to come (IPCC, Stocker et al. (2014)). A major decline in Arctic sea ice is already observed (Kwok and Rothrock, 2009; Comiso et al., 2008) and climate change projections suggest that, due to rising temperatures, a complete disappearance of summer sea ice could occur as soon as 2050 (Overland and Wang, 2013). Obtaining an improved understanding of the changing Arctic Ocean, its transport properties of heat, freshwater as well as carbon and nutrients, and its interaction with sea ice and the overlying atmosphere is therefore of utmost importance.

Despite recent improvements in observing capabilities (Lee et al., 2010), the Arctic Ocean remains one of the least explored areas of the World Ocean. This is due to the harsh environmental conditions of the region, but also due to logistical and political difficulties in maintaining sustained Arctic-wide, ideally autonomous, ocean observations. Fortunately, many polar-orbiting satellites obtain important ocean and sea ice parameters over the sub-Arctic region, such as sea surface height (SSH), sea surface temperature (SST), ocean color and sea surface salinity (SSS). However, over sea ice covered regions satellite measurements of the ocean surface are limited. To enhance our insight into the Arctic environment a joint analysis of observational efforts is therefore required. However, to understand large scale circulation processes in the Arctic Ocean the community will have to rely on numerical ocean circulation models due to the continued substantial under-sampling of the Arctic under sea ice cover.

The representation of the Arctic Ocean circulation in existing ocean models considerably improved during the last 10 years, to the point that today many models reasonably well reproduce variability of SSH (Koldunov et al., 2014), while for the components of the freshwater balance the picture is mixed (Jahn et al., 2012) and for circulation and water masses models show significant discrepancies (Proshutinsky et al., 2011).

One method to further increase the resemblance between models and available observations is data assimilation. The models with data assimilation can be used to draw conclusions about variations in Arctic Ocean parameters on decadal scales, and to reveal mechanisms which drive changes in Arctic circulation.

Stammer et al. (2016) described the state of ocean data assimilation in the context of climate research. As described there, ocean data assimilation became a mature field for the ice-free ocean. However, assimilation in coupled ocean-sea ice or fully coupled climate models is still at its infancy and needs considerable attention. This also includes the use of sea ice parameters to constrain coupled ocean-sea ice models and to understand the coupling between sea ice and the underlying ocean and the atmosphere.

Chevallier et al. (2016) recently reported results from the ORA-IP inter-comparison project for Arctic sea ice parameters using global ocean-sea ice reanalyses with and without assimilation of sea ice data. They found good agreement in the reconstructed concentration but a large spread in sea ice thickness due to biases related to the sea ice model components.

The approaches to the sea ice assimilation are similar to the way ocean variables are assimilated in ocean models and range from nudging (e.g. Lindsay and Zhang (2006); Tietsche et al. (2013)) to the use of ensemble Kalman filter ( e.g. Lisæter et al. (2003); Xie et al. (2016)). The sea ice sensitivity study of Koldunov et al. (2013) was among the first prerequisites to a full data assimilation attempt in the Arctic with the adjoint method. The authors looked at the sensitivity of sea ice parameters to external atmospheric forcing parameters (see also Kauker et al. (2009)). The former study revealed the impact of spring atmospheric temperatures on summer sea ice concentration and extent. The study of Kauker et al. (2009) underlined that wind stress changes are important for changing summer sea ice thickness.

More recently, Fenty et al. (2015) studied the impact of assimilating sea ice concentration (and ocean) data into a global, eddy permitting ocean circulation model using the adjoint method. In that study the circulation for the year 2004 was reconstructed. By comparing a setup with and without assimilation of sea ice concentration, the authors demonstrate that sea ice concentration data reduce model misfits in the Arctic with respect to upper ocean stratification and reduces ICESat-derived Arctic ice thickness errors.

The present study builds on the work of Fenty et al. (2015) and advances it by performing a multi-year data assimilation for the coupled Arctic Ocean-sea ice system. To be computationally feasible, the study is based on a regional Arctic configuration, nested laterally into a North Atlantic-Arctic solution (Serra et al., 2010). The goal of the study is to investigate the changes in the Arctic during the period 2000 - 2008. This period is characterized by significant changes in the Arctic Ocean and by increased amounts of Arctic observations. This makes it a good test period for the assimilation system and can provide first scientific applications. At the same time, the consistency of the assimilated EUMETSAT sea ice data (OSI-SAF, 2015) with the used sea ice model is being tested, as are its impact on the estimate of the ocean circulation and unobserved ice parameters such as sea ice thickness.

The remaining paper is structured as follows: after an introduction to the model configuration and the assimilation method in Section 2, the impact of the assimilation on the sea ice concentration is discussed in Section 3. Section 4 focuses on how the sea ice state is adjusted by changing the control variables and Section 5 summarizes the impact on the ocean state and the sea ice thickness. Concluding remarks follow in Section 6.

## 2 Methods

Our study is based on a regional configuration of the MITgcm coupled ocean-sea ice model (Marshall et al., 1997) and the respective ECCO adjoint framework. The model set-up, the data assimilation and the optimization results are described in the following subsections.

### 2.1 Model set-up

The model domain covers the northern North Atlantic and the Arctic Ocean (Fig. 1) with the model grid being curvilinear and a subset of the 16-km resolution Atlantic-Arctic model (ATL06) reported in (Serra et al., 2010). The model uses z-coordinates and has 50 levels, with resolution varying from 10 meters in the top layers of the water column to 550 meters in the deep parts of the ocean. The bathymetry is based on the ETOPO2 database (Smith, 1997) with no artificial deepening or widening of the Nordic Seas passages being applied.

As atmospheric forcing, the model uses the atmospheric state from the 6-hourly NCEP R1 reanalysis (Kalnay et al., 1996), including 2-meters air temperature, precipitation rate, 2-meters specific humidity, downward shortwave radiation flux, net shortwave radiation flux, downward longwave radiation flux, 10-meters zonal wind component and 10-meters meridional wind component. The surface fluxes of heat, freshwater and momentum are derived via bulk formulas. At the open southern boundary, roughly at 48° N in the Atlantic, results from a 60-year long integration of the ATL06 model are used. The ATL06 was in turn forced laterally at 33° S by a 1° resolution global solution of the MITgcm forced by the same NCEP data set (see (Serra et al., 2010) for details). At the northern boundary a barotropic net inflow of 0.9 Sv into the Arctic is prescribed at Bering Strait, balancing the corresponding outflow through the southern boundary. An annual averaged river run-off (Fekete et al., 1999) is applied in the North Atlantic, while seasonally varying run-off is used for the Arctic rivers.

The MITgcm offers a wide variety of modules that can simulate different aspects of the unresolved ocean physics. For the vertical mixing parameterization we use the K-Profile Parameterization (KPP) scheme of (Large et al., 1994). The model is operated in a hydrostatic configuration with an implicit free surface. The sea ice component is based on a Hibler-type (Hibler, 1979, 1980) viscous-plastic dynamic-thermodynamic sea ice model. The thermodynamic part of the model is the so-called zero-layer formulation following Semtner (1976) with snow cover as in Zhang et al. (1998). The temperature profile in the ice is assumed to be linear, with constant ice conductivity. Such a formulation implies that the sea ice does not store heat, and, as a result, the seasonal variability of sea ice is exaggerated (Semtner, 1984). To reduce this effect we use the sub-grid scale heat flux parameterization following Hibler (1984). Moreover, we use the viscous-plastic rheology scheme of Hibler (1979) with an extended line successive over-relaxation (LSOR) method (Zhang and Hibler, 1997). A comparison of the effect of different

rheology schemes in MITgcm is provided by Losch et al. (2010). Recently, Nguyen et al. (2011) applied the coupled MITgcm in a regional Arctic Ocean study and reported values for many model parameters used in our study.

## 2.2 Adjoint data assimilation approach

Similar to the work of Fenty et al. (2015), our assimilation also employs the ECCO adjoint methodology to bring the coupled sea ice-ocean general circulation model into consistency with assimilated data and prior uncertainties. The particular implementation used here builds on the set-up of the GECCO2 synthesis (Köhl, 2015) but was extended to facilitate the additional assimilation of sea ice parameters. A complete list of parameters assimilated and their sources are presented in Table 1. The collection of hydrographic observational data in the Arctic Ocean used in the present work is not comprehensive and does not include, for example, ice-tethered profiler data (Toole et al., 2011; Krishfield et al., 2008). In the present pilot study we decided to stick to two well-structured data sets available at the time we have started our efforts.

While using the adjoint method, an uncertainty-weighted sum of squares of model-data misfits is minimized in an iterative fashion using the gradient of the cost function with respect to a number of control variables. The cost function $J$ is defined as follows:

$$J = \sum_{t=1}^{t_f} [y(t) - E(t)x(t)]^T R(t)^{-1} [y(t) - E(t)x(t)] +$$

$$v^T P(0)^{-1} v + u_m^T Q_m^{-1} u_m + \sum_{t=0}^{t_f - 1} u_a(t)^T Q_a(t)^{-1} u_a(t) \quad (1)$$

where $y(t)$ is a vector of assimilated data in time $t$, $x(t)$ is a vector of the model state, $E(t)$ is a matrix which maps the model state to the assimilated data, $v$ is the difference between the first guess initial condition and the model state at the beginning of the assimilation period (only for the first year), $u_m$ is the difference between the first guess time mean atmospheric state and the optimized mean atmospheric state, $u_a(t)$ is the difference between the first guess time-varying atmospheric state and the optimized time-varying atmospheric state. Additional weights $R(t)^{-1}$, $P(0)^{-1}$, $Q_m^{-1}$ and $Q_a(t)^{-1}$ control the relative contribution of different terms in the cost function. More detailed description of the cost function and optimization procedure can be found in Fenty et al. (2015).

The MITgcm is suitable for the automatic generation of adjoint code by the Transformation of Algorithms in FORTRAN (TAF) source-to-source translator (Giering and Kaminski, 1998; Giering et al., 2005). Koldunov et al. (2013) used the MITgcm and its adjoint to perform an analysis of the Arctic-wide adjoint-based sea ice sensitivities to atmospheric forcing.

Here we use a version of the MITgcm with an improved adjoint of a thermodynamic ice model (Fenty and Heimbach, 2013a, b). The adjoint model was modified here similarly to Köhl and Stammer (2008) to exclude KPP module and increase diffusivity values compared to the forward run. This is done to avoid exponentially growing adjoint variables. The sea ice module was active in the adjoint integration, but the part of the sea ice dynamics which treats rheology was switched off, so that the sea ice model was in a free drift configuration. This approach led to a reduced (approximate) adjoint producing smoother adjoint

gradients. These gradients can still be successfully used to improve the large scale state of the model (see Köhl and Willebrand (2002) and Köhl and Stammer (2008) for more details). Similar simplifications of the adjoint model were used by Fenty et al. (2015) and Liu et al. (2012) provided an evaluation of the effect of modifications in the parameterizations on the adjoint. They confirm mostly small changes, although regionally some patterns of the gradients may shift. Since the gradients are only a
means to find the cost function minimum and the forward code (and thus the minimum itself) is unmodified, changes to the gradient may lead to lower performance in finding the minimum but not to different states once the minimum is found.

In contrast to Köhl (2015), additional control variables are optimized and the frequency of the updates is enhanced to once per 3 days in order to reflect shorter time scales of sea ice variability. The final list of control variables is: surface (2m) air temperature, surface (2m) specific humidity, surface (10m) zonal and meridional wind velocity, precipitation rate, downward
shortwave radiation, and initial temperature and salinity for the first year of assimilation. For the atmospheric control variables, uncertainties were specified as the maximum of the standard deviation of the NCEP fields for 1948-2008 time period and the errors for the mean components of air temperature, humidity, precipitation, downward shortwave radiation and wind were specified as $1^\circ$ C, 0.001 kg/kg, 1.5 x $10^{-8}$ mm/s, 20 W/m$^2$ and 2 m/s, respectively. For the downward shortwave radiation both mean and time varying parts were set to 20 W/m$^2$.

We employ the same uncertainty weights for hydrographic and satellite data as Köhl (2015), while for sea ice concentration we specify a constant error of 50%. We verified the sensitivity of our results by using space-time varying sea ice uncertainty estimates as they became available, as well as different values of a constant error. Results of the sea ice assimilation with variable uncertainties were very similar to the ones with a constant error value of 50%.

The data assimilation is performed in one year chunks. The use of one year segments is related to technical reasons; we are
not able to get useful sensitivities for the time period longer than a year for all years of our 2000-2008 assimilation period. We were successful in completing a 2-year assimilation at one occasion (2005-2004), but the results for sea ice area and thickness were not noticeably different from the 1-year chunk assimilation.

Each of the iterative cost function reductions is performed until the cost function differs by less than 1% in two consecutive iterations. The cost is dominated by SIC and SST data, which easily respond to the surface controls, and the adjoint method
quickly reduced the misfits of those data, so that the number of iterations was usually less than five (it is 3 iterations for 2000, 2003, 2004, 2005, 2006 and 2007, 4 iterations for 2002 and 2008 and 5 iterations for 2001). After the first year assimilation, we move to the next year using the final state of the previous year's successful iteration as initial conditions. Therefore, the iteration termed 0 in the following makes already use of an improved initial condition from the assimilation in the previous year, and is thus not equivalent to a free run starting from climatology. For the impact on the ocean circulation, we consider
also the free run to demonstrate the impact of changing the initial conditions by assimilating data during the preceding year.

Fig. 2 shows the percentage decrease in model-data differences. The red color indicates reduction in total model-data difference (FC), while other colors indicate the reduction of the differences for individual variables. Negative values mean that there is an increase in model-data difference for that variable.

The largest total reduction (about 16%) is obtained for the year 2008, while the smallest (about 2%) is obtained for the year
2005. The average reduction for all years is about 9%. The strongest cost reductions for individual variables is obtained for

the sea surface temperature (SST) and sea ice area (SIA), with an overall average of about 23% and 26%, respectively. The least successful cost reduction is obtained for the mean dynamic topography (MDT), with many years in which the model-data differences for this variable slightly increased. In 2004 the cost reduction of sea ice area was about 30%, less than that reported by Fenty et al. (2015) (49%), which may partly be explained by differences in the first guess solution.

Taking into account differences in the amount of sea ice concentration and sea surface temperature data compared to the amount of hydrography data, it is not surprising that most of the contributions to the total reduction of the cost function are from SIC and SST. Hence most of the improvements can be expected to happen in these fields, while changes in the state of the ocean is expected to be small.

    In the following we concentrate mainly on results related to changes of the sea ice conditions, with only a brief discussion
of ocean state changes later on.

## 3   Sea ice concentration changes

Fig. 3 shows in the top two rows the sea ice concentration for the winter time period (March of the year 2005) from satellite and from model runs, before and after data assimilation, together with the changes of the latter two relative to observations. Since most of the Arctic Ocean is covered by sea ice with high concentrations, the largest improvements are in the position of
the ice edge. Most noticeable is the decrease in the SIC along the east coast of Greenland after data assimilation. During the initial run of the model, there is a tongue of the sea ice extending towards the open ocean. After data assimilation the tongue did not disappear completely, however, it declined considerably.

    During the summer period (September 2005), shown in the bottom two rows of Fig. 3, there are improvements both in the sea ice edge and in the SIC of the interior sea ice field. Initially, the sea ice edge was not very far from observations, but after
data assimilation the match between model and data is improved. The SIC in the central parts of the Arctic Ocean increased and became closer to the satellite data. A direct comparison to the results by Fenty et al. (2015) is hindered by the fact that differences less than 15% are blanked out in their study and by the different years analyzed.

    In order to test the consistency of the estimate with the observations and the uncertainties we compare the spatial distribution of monthly mean sea ice concentration absolute differences before and after data assimilation to the maps of spatial distribution
of monthly mean total standard error in the ESA SICCI sea ice concentration product ((ESA SICCI, 2013)). The latter provide daily spatially varying estimates of sea ice concentration errors. The absolute differences after assimilation correspond well to the total standard error spatially and by value with only few spots along the edges with very high absolute differences (not shown).

    In contrast to 2005, identifying changes in the SIC for March 2007 (Fig. 4) is more challenging. Practically all the differences
between simulations and satellite data are along the ice edge and there seems to be not much change between the initial state of the model and the state after assimilation. For example, the noticeable negative anomaly around Franz Joseph Land is not developed further after SIC assimilation. This particular negative SIC anomaly is most probably dynamical in nature, and can not be handled properly by the simplified ice dynamics scheme (free drift) used in the adjoint model to calculate changes of

the model parameters. The spatial distribution of SIC during September 2007 (Fig. 4) already bears a good resemblance to the satellite data before the assimilation. Improvements are mostly visible in the central parts of the Arctic Ocean, where the too low SIC is increased. The ice edge also became closer to observations, but the amount of sea ice in the Amerasian basin remains larger compared to observations. In this region the SIC in the unconstrained run is high (with also thicker sea ice),
which is not easy to remove by thermodynamic corrections of the forcing and, due to the high SIC and thickness, not easy to move by changes in wind forcing. This possibly indicates some limitations of the approach, where the corrections mostly come from the thermodynamic forcing and the assimilation period is short.

The seasonal cycle of sea ice area (SIA) and sea ice extent (SIE) are shown in Fig. 5, again for years 2005 and 2007. Results for SIA for both years show that values of SIA in general are getting closer to satellite observations as a result of the
SIC assimilation. One would expect that, close to the beginning of the assimilation period (1st of January), corrections of the atmospheric forcing did not have enough time to considerably influence sea ice parameters. This is true for SIA in 2007, when sizable differences between initial and last iterations only first appear in May. However, SIA in 2005 gets considerably closer to observations already in February, indicating that atmospheric corrections actually can affect sea ice parameters relatively fast even during winter.
For both years, SIA shows overall improvement during the whole year; but this is not the case for the SIE. In 2005 the SIE good match between initial iteration and satellite data during summer months disappears after assimilation, with considerable underestimation of SIE. In 2007 there is an overall SIE improvement after the assimilation, but there are again months with a considerable SIE underestimation. Both metrics suffer from the inability to guarantee that improvements in this metric also lead to an overall improved match in the spatial sea ice coverage, since a perfect total SIA or SIE evolution may still correspond
to considerable differences to the data in their regional distribution. Chances of having SIE distribution close to observations with quite different spatial shape of the sea ice field are very high. This calls for changing the common practice of model evaluation by only comparing their ability to simulate present day SIE without considering the sea ice spatial distribution (e.g. Dukhovskoy et al. (2015)).

With respect to the model performance, two better metrics are the sum of absolute differences (SoAD) for SIA and SIE,
which at least to some extent consider differences in spatial distribution by penalizing positive and negative differences at every grid point. Monthly values of the SIA SoAD before assimilation, after assimilation and the respective differences between the two (in percent) are shown in Fig. 6. Before assimilation, largest SoAD appear during summer months ($> 2 \text{x} 10^6$ km$^2$), while in other seasons they are about $1.5 \text{x} 10^6$ km$^2$. Interesting to note, values of SoAD in March and September are quite similar, despite the large differences in ice cover in the two months. One of the possible reasons is that location of the ice edge in
those extreme months is relatively stable compared to spring and fall when the ice pack is contracting and expanding. After the assimilation the most notable improvements also occur for summer months, but with the addition of September. After the assimilation, March values show only about 10% improvement, while September values have about 25% improvement on average. There is no clear indication that assimilation of SIC on the yearly basis gradually improves the simulated sea ice, due to, for instance, better initial conditions in January. For some months the decrease in SIA SoAD after assimilation can be as
little as 1%, although it is always getting smaller. The same is not the case for the SIE SoAD.

As expected, SIE SoAD values (Fig. 7) are larger, with a maximum in summer and September before the data assimilation. Assimilation is most effective for a reduction of SIE SoAD in September (about 25% on average). After the assimilation October becomes, in addition to summer months, one of the months with relatively large SIE SoAD differences. October is also a month when (during 5 out of 9 years) after assimilation the SIE SoAD increased. The SIE SoAD, similarly to the SIA SoAD, do not show any obvious tendency from the first year to the last.

## 4 Control variables

As mentioned in Section 2.2, the model is brought into consistency with observations by adjusting a number of control variables. The strength and spatial distribution of the adjustments carry important information about the way the optimization procedure changes the forcing and the initial conditions in order to bring the state of the model closer to the observed state. Figure 8 shows the area-mean temporal variation of the corrections to several control variables over the year 2005 in absolute values and normalized by the uncertainties. Also shown are the spatial distribution of the corrections for the month when their strength is at its maximum.

As expected, there are strong changes in the surface atmospheric temperature. Its modification is probably the easiest way to change the sea ice concentration by increasing temperature when/where a reduction of SIC is required and vice-versa. The spatial distribution of corrections in 2005 (Fig. 8, top row, left column) compares very well to the difference between first guess and satellite SIC data in the central Arctic (Fig. 3). In order to increase SIC in the Eurasian Basin, the optimization reduces the surface atmospheric temperature in June by about 2 degrees in this region on average, reaching 3 degrees in some places. Positive surface atmospheric temperature (SAT) corrections over the Arctic shelf seas helps to reduce extra sea ice generated there by the model during summer months (not shown).

The corrections to the downward shortwave radiation (Fig. 8, second row) show temporal variations and a spatial distribution similar to the SAT corrections, but the magnitudes are quite small. Corrections to the zonal and meridional wind components (Fig. 8, third and last rows) are on average quite small in absolute values, but locally can reach 10 m/s. The wind corrections are mainly concentrated along the shore and summer ice edge and, contrary to the SAT corrections, it is difficult to associate them to some particular large-scale sea ice change.

Dimensional values of the corrections do not directly provide information about the relative importance of changes in the controls for bringing the model into consistency with observations. However, due to the relatively small number of iterations, we can use values of the corrections normalized by uncertainties as a reasonable measure of the relative importance of changes in control parameters. Spatial distributions and monthly means of absolute values of normalized corrections for the year 2005 are shown in Fig. 8.

Wind corrections seem to play integrally a larger role, with a maximum in May. This agrees well with results of (Kauker et al., 2009), who used an adjoint sensitivity analysis to determine the relative contribution of different atmospheric and ocean fields to the September 2007 sea ice minimum and found that the May-June wind conditions are one of the main factors in setting up extremely low sea ice conditions in Summer 2007. The maximum contribution of air temperature corrections occurs in June

and it is about a factor of five smaller than the contribution of the wind corrections. However, using free drift in the adjoint biases the sensitivities towards larger sensitivities of sea ice to wind changes. Since measuring the impact by the normalized corrections relies on the assumption of correct sensitivities, the results may be also biased to too large an impact by the wind.

Given the absence of proper sea ice dynamics in the adjoint model (only free drift is used) and lack of many important processes in the forward model (such as tides or waves), the question remains to what extent corrections to control variables reflect deficiencies in the forcing fields or a compensation to the sea ice model or sea ice data deficiencies, particularly since in the Arctic the NCEP reanalysis seems to perform well near the surface (Jakobson et al., 2012). For example, temperatures decreasing over areas with high SIC during summer months in order to grow ice and temperatures increasing over low SIC areas, could be an attempt of the assimilation system to fix problems associated with the sea ice movement. But it could equally also point to problems of the correct attribution of sea ice concentrations from satellite data. In both cases, corrections to atmospheric control variables will not improve the quality of the original atmospheric forcing, but on the contrary may make it worse.

## 5   Improvements in sea ice thickness and ocean state

The adjoint assimilation leads to dynamically consistent model solutions, which along with directly assimilated variables may considerably improve variables of the simulation for which no observations are available. In case of SIC assimilation, one obvious candidate for improvement is the sea ice thickness (SIT). We also consider changes in the ocean state which result from the combined effect of assimilating ocean parameters and indirectly of the SIC assimilation, due to the coupled nature of the assimilation procedure and the forward model.

### 5.1   Sea ice thickness

Changes in SIT as a result of SIC assimilation and comparisons of the former with satellite data are shown in Fig. 9. The satellite ice thickness data are obtained from ICESat campaigns (Kwok et al., 2007), distributed on a 25-km grid and available from the NASA Jet Propulsion Laboratory (http://rkwok.jpl.nasa.gov/icesat/index.html). ICESat sea ice thickness estimates are considerably larger than those in the simulations, especially in the Canadian sector of the Arctic Ocean. One should note that the uncertainty for this observational data is quite large (just better than 0.7 m, Kwok et al. (2007)), while the spatial distribution of the thickness is probably realistic (Kwok and Cunningham, 2008).

The ice in October-November during 2005 became thicker in the Eurasian Basin of the Arctic Ocean after assimilation and in general became closer to the observed thickness distribution. The thickness increase is considerable, reaching 0.5 m in some places. The shape of the region with the largest thickness increase in the Eurasian Basin resembles the shape of the September SIC distribution (Fig. 3) and because of its similarity in pattern it is probably a result of the control variable's corrections that aim to thermodynamically increase SIC in this region. Results for October-November 2007 are similar, with improved thickness along the continental shelf of the Eurasian Basin. However, thickness increase is not as strong as for 2005, reaching only about 0.3 m. A general tendency of these improvements is an increase in thickness in the central Arctic and the Canadian

Basin, while regions with thin ice over the shelf seas tend to decrease in thickness. This tendency was also shown by Fenty et al. (2015) for the year 2004.

To summarize, the visual comparison with available satellite data hint to a general improvement of the SIT spatial distribution.

## 5.2 Ocean changes

Local changes of the SIC are caused by corrected atmospheric conditions (see above), which in the coupled system will also affect near-surface ocean parameters. To some extent changes can also come about through change in the ocean circulation and we want to investigate therefore how large those changes are and to what extent they could contribute to the sea ice improvements.

Fig. 10 shows differences in temperature and salinity between the initial and final iterations of the assimilation system for June and September of year 2005. The month of June is chosen because corrections to thermodynamic control variables during this month are largest (see above in Section 4). The sea surface temperature differences are mostly positive along the ice edge, where the model produces too much ice in the initial iteration (Fig. 3), and lower in magnitude in the central part of the Arctic Ocean. In June, considerable temperature differences cover a much smaller area compared to September, since most of the shelf seas are still covered by high concentrations of sea ice and most of the additional energy resulting from the correction to thermodynamic control variables is spent directly in the sea ice melting.

The surface salinity (Fig. 10, right column) shows an increase in the Eurasian Basin, caused by additional sea ice production (or less melting). There is a decrease of salinity around the sea ice edge due to melting of excessive sea ice formed in the initial iteration. In September, however, there is a pronounced increase in salinity in most of the Arctic shelf seas. This might be a result of the local increase in sea ice production in areas which become free of ice due to the summer corrections (e.g. Laptev Sea), but still have quite negative temperatures in the original forcing which are not corrected in September (corrections in September are quite small) at the onset of the freezing period.

Due to the relatively short assimilation periods (1 year) and to the extremely low amount of vertical temperature/salinity profile observations, improvements in the vertical distribution of temperature and salinity after 9 years of assimilation are quite small. Nevertheless, the positive bias in the Atlantic Water layer temperature of the Eurasian Basin, which is characteristic for the forward run, has been slightly reduced (not shown). On the other hand, changes in the upper part of the water column due to sea ice corrections, although hardly penetrating deeper than the first 50 meters, may influence integral fluxes at the borders of the Arctic Ocean.

We have calculated volume, heat and freshwater fluxes (Table 2) through the main passages of the Arctic Ocean (except for Bering Strait, where fluxes are largely prescribed in the model by the boundary conditions). Along with the initial and final iterations, results for a no-assimilation forward simulation were analyzed in order to remove the effect of changing the initial conditions at the beginning of each assimilation year. These may lead to changes of long-term variability and may affect the fluxes towards the end of the assimilation period. We also show mean fluxes for August-September of year 2005 and compare

them to the results of Tsubouchi et al. (2012), who applied an inverse model to data obtained in summer 2005 to calculate net fluxes of volume, heat and freshwater around the Arctic Ocean boundary.

Differences in the total mean volume flux are quite small for all passages. This is probably due to the fact that the volume flux is mostly controlled by the wind stress, which means that the corrections of the control variables discussed above do not contribute considerably to changes in the ocean circulation. This is expected since the amount of sea ice concentration data is much larger than the number of hydrographic observations in the Arctic Ocean, so that the assimilation system tries to change control variables in a way that will have larger impact on the sea ice. However, episodically, significant changes can be observed (for example in summer 2008) when modifications in the throughflows across Fram Strait are noticed, which are about 60% larger than in the forward simulation without data assimilation (Fig. 11a).

Differences in the heat flux (Fig. 11b) at Fram and Davis Straits can be episodically relatively large, but they do not show any particular tendency and may be related to the local heating or cooling in the vicinity of the sections. Table 2 summarizes the mean differences for the analyzed passages and, although hardly visible in the time series (not shown), heat flux differences for the St. Anna Trough are the largest on average, reducing the heat export from the Arctic Ocean by about 80%.

The freshwater flux differences (Fig. 11c) are most visible in the Fram Strait time series, but positive and negative differences remain comparable to the forward run and compensate each other, such that on average the relative difference is only about 3%. Large relative differences again occur for the St. Anna Trough (Table 2), which is located in an area with strong atmospheric corrections during most of the years.

Considering Tsubouchi et al. (2012) to be a good approximation of observed values in August-September 2005, it is hard to definitely conclude if ocean fluxes become better or worse after the assimilation (Table 2). Some values, such as the volume flux through Davis Strait and the Barents Sea Opening, or the freshwater flux in the Fram and Davis Straits, have changed and became closer to the values of Tsubouchi et al. (2012). Other values moved even further away from their estimates.

From the combined analysis of Fig. 11 and Table 2 one can conclude that, while on average most of the transports are hardly affected by the assimilation, during some periods relative large differences between the simulations with assimilation and the forward run without assimilation can be seen and may reach 60-100% for major straits.

## 6   Concluding remarks

Results from a multi-year data assimilation attempt based on a coupled Arctic Ocean-sea ice system were presented. The largest improvements relative to simulations without data assimilation were seen for the sea ice concentration (SIC) and sea surface temperature. Most of the improvements in the SIC happened during summer months and manifest themselves in a more realistic position of the sea ice edge and in SIC values closer to observations in the central Arctic.

The seasonal cycle of the monthly mean sea ice area (SIA) shows an overall improvement after assimilation, while sea ice extent (SIE) becomes worse during some months. The later fact demonstrates that the total mean SIE and SIA are not good measures for the model success in simulating sea ice, particularly considering the obvious improvements in spatial sea

ice distribution. In order to obtain more meaningful estimates of the sea ice improvements, we consider sums of absolute differences (SoAD) for SIA and SIE. The largest reduction of the SoAD happened during the summer months.

An obvious suggestion for improving the sea ice estimation is to consider larger assimilation periods or even best to use a single assimilation window. By this, data from later years may influence the corrections and the state of all preceding years. However, a long memory of the system seems to be not very evident in the assimilation. We have assimilated data in yearly chunks and one could expect that SoAD between observations and initial simulations (before assimilation) would gradually improve due to better initial conditions, at least over the first few years. However, we do not observe this effect in our experiments.

The comparison to available but limited sea ice thickness observations shows that SIC assimilation reveals some improvements in sea ice thickness (SIT), despite these observations not being directly assimilated. The amount of assimilated ocean observations in the water column of the Arctic Ocean is almost negligible compared to the amount of SIC data. However, the ocean state is affected indirectly by SIC assimilation, for example due to the freshwater fluxes related to the additional melting or freezing and by changes in the ocean exposure to the atmosphere caused by changes in SIC. The transports of ocean properties do not change on average after the assimilation, but episodically they can be quite different from the corresponding transports in simulations without assimilation. The latter can still be important for local process studies or model validation against observations that are limited in time.

With the use of the adjoint assimilation technique, we produced a model simulation that is considerably closer to observations and at the same time dynamically consistent. This data can be used for further understanding of the reasons and consequences of changes in the Arctic Ocean.

*Acknowledgements.* This work was funded in part by the European Union $7^{th}$ framework program through the MONARCH-A Collaborative Project, FP7-Space-2009-1 contract No. 242446. NK is also supported by the project S1 ("Climate models as metrics") of the Collaborative Research Centre TRR 181 "Energy Transfer in Atmosphere and Ocean" funded by the German Research Foundation (DFG). The model integrations were performed at the Deutsches Klimarechenzentrum (DKRZ), Hamburg, Germany. We thank two anonymous reviewers for their constructive comments, which helped to improve the manuscript.

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

**Table 1.** Datasets used in the assimilation procedure.

| Dataset | Source |
| --- | --- |
| Mean Dynamic Topography | MDT from Technical University of Denmark (Knudsen et al., 2011; Knudsen and Andersen, 2013; Cheng et al., 2014) |
| Monthly SST | Remote Sensing Systems [CIT] |
| Sea Level Anomalies | TOPEX/Poseidon, ERS-1,2 and Envisat, AVISO [CIT] |
| EN3 hydrographic data | Ingleby and Huddleston (2007) |
| NISE hydrographic data | Nilsen et al. (2008) |
| Sea ice concentration | OSI-SAF (2015) |

**Table 2.** Mean values of different fluxes through Arctic Ocean passages.

| Parameter and passage | Forward | After assimilation | Difference in % | Forward 2005 | After assimilation 2005 | Tsubouchi et al., 2012 |
|---|---|---|---|---|---|---|
| **Volume flux (Sv)** | | | | | | |
| Fram St. | -3.12 | -3.12 | -0.02 | -4.0 | -4.49 | -1.6 ± 3.9 |
| Davis St. | -0.50 | -0.55 | 4.72 | 0.44 | 0.03 | -3.1 ± 0.7 |
| Barents Sea Op. | 2.78 | 2.81 | 0.88 | 3.5 | 3.6 | 3.6 ± 1.1 |
| St. Anna Tr. | -2.01 | -2.01 | 0.18 | | | |
| **Heat flux (TW)** | | | | | | |
| Fram St. | 38.76 | 38.62 | -0.36 | 41.5 | 39.9 | 62 ± 17 |
| Davis St. | 7.94 | 7.69 | -3.12 | 8.6 | 6.3 | 28 ± 3 |
| Barents Sea Op. | 83.10 | 84.07 | 1.17 | 111.8 | 115.8 | 86 ± 19 |
| St. Anna Tr. | 1.02 | 0.20 | -80.13 | | | |
| **Freshwater flux (mSv)** | | | | | | |
| Fram St. | -113.50 | -109.80 | -3.20 | -173.0 | -141.0 | -70.0 ± 40 |
| Davis St. | -25.60 | -27.27 | 6.50 | 13.5 | -11.3 | -119 ± 14 |
| Barents Sea Op. | -21.81 | -22.37 | 2.57 | -22.5 | -22.0 | -31 ± 13 |
| St. Anna Tr. | 6.84 | 8.44 | 23.32 | | | |

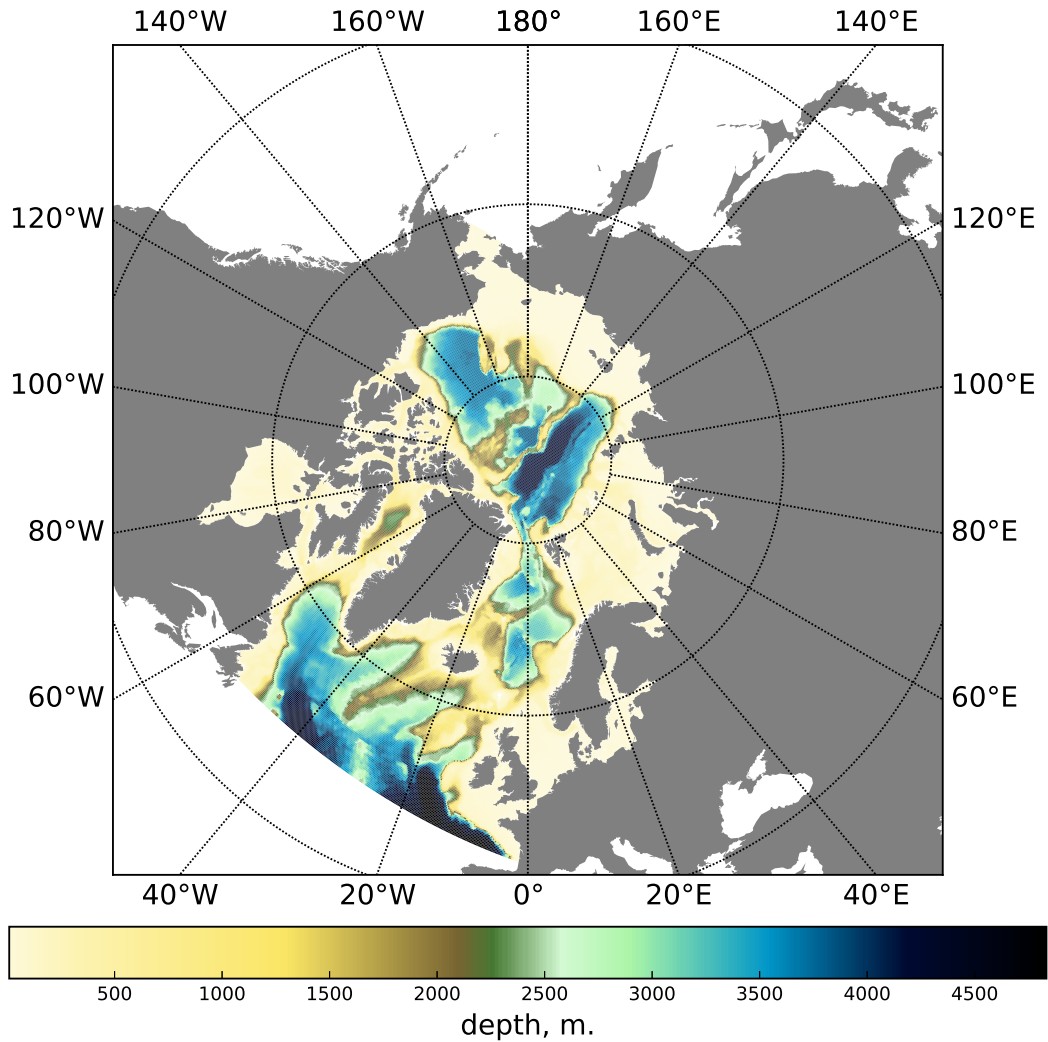

**Figure 1.** Model domain with bathymetry.

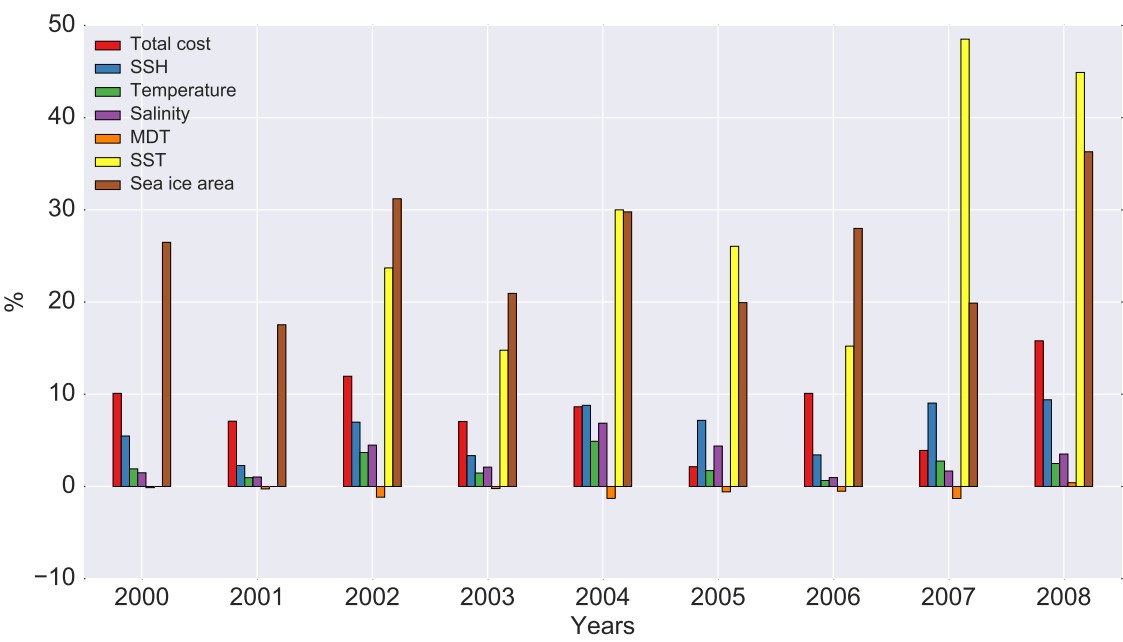

**Figure 2.** Total cost reduction and individual contributions to the reduction from different assimilated variables. During the first two years SST assimilation is not performed (no data).

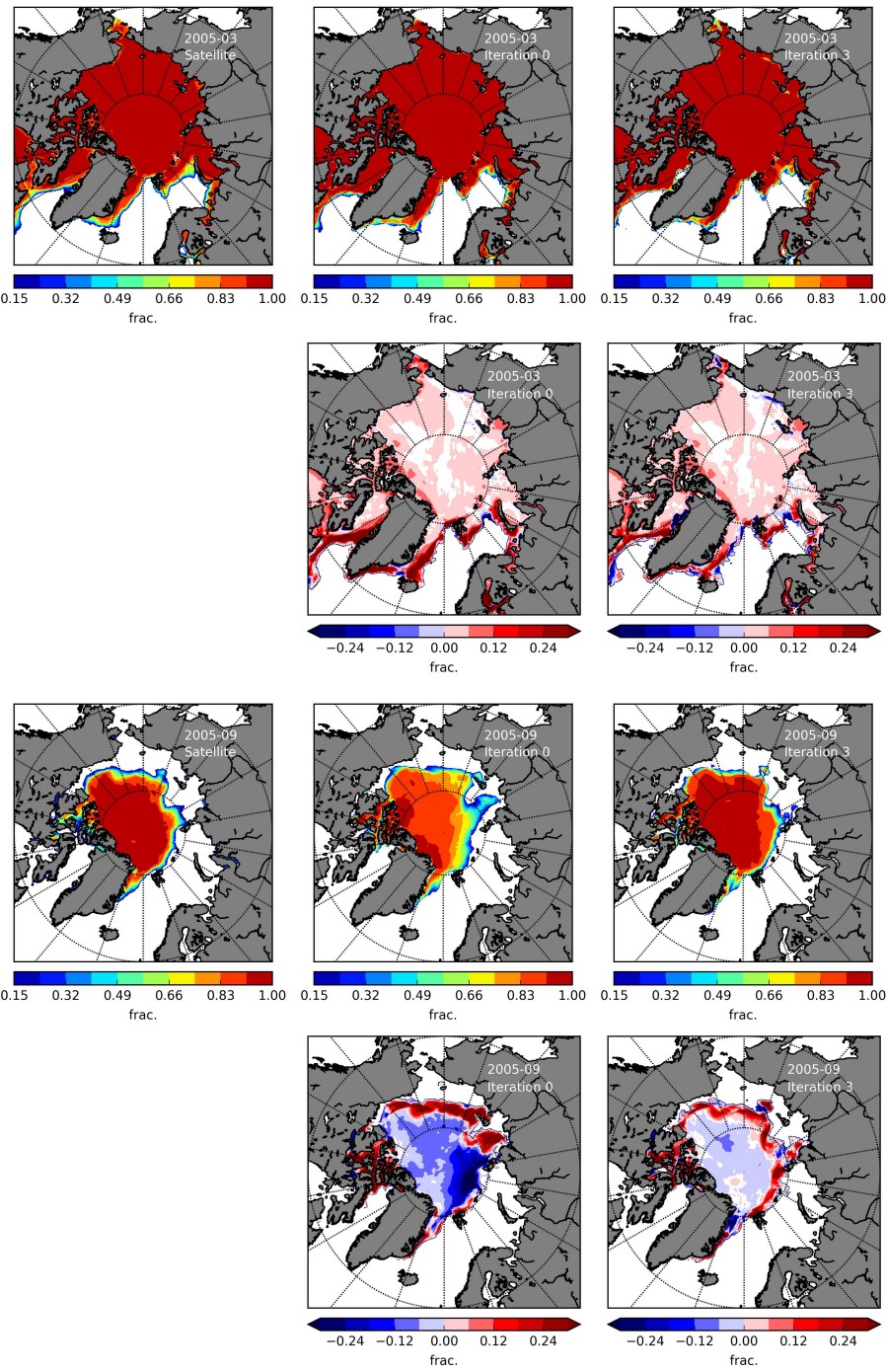

**Figure 3.** Spatial distribution of sea ice concentration (SIC) for the year 2005 (year of the local sea ice maximum) during March (first row) and September (third row). Assimilated satellite data (left column), model results from the run without corrections (middle column) and model results during the last assimilation iteration (right column) are shown. The second and fourth rows correspond to the differences between the model solutions and the observations.

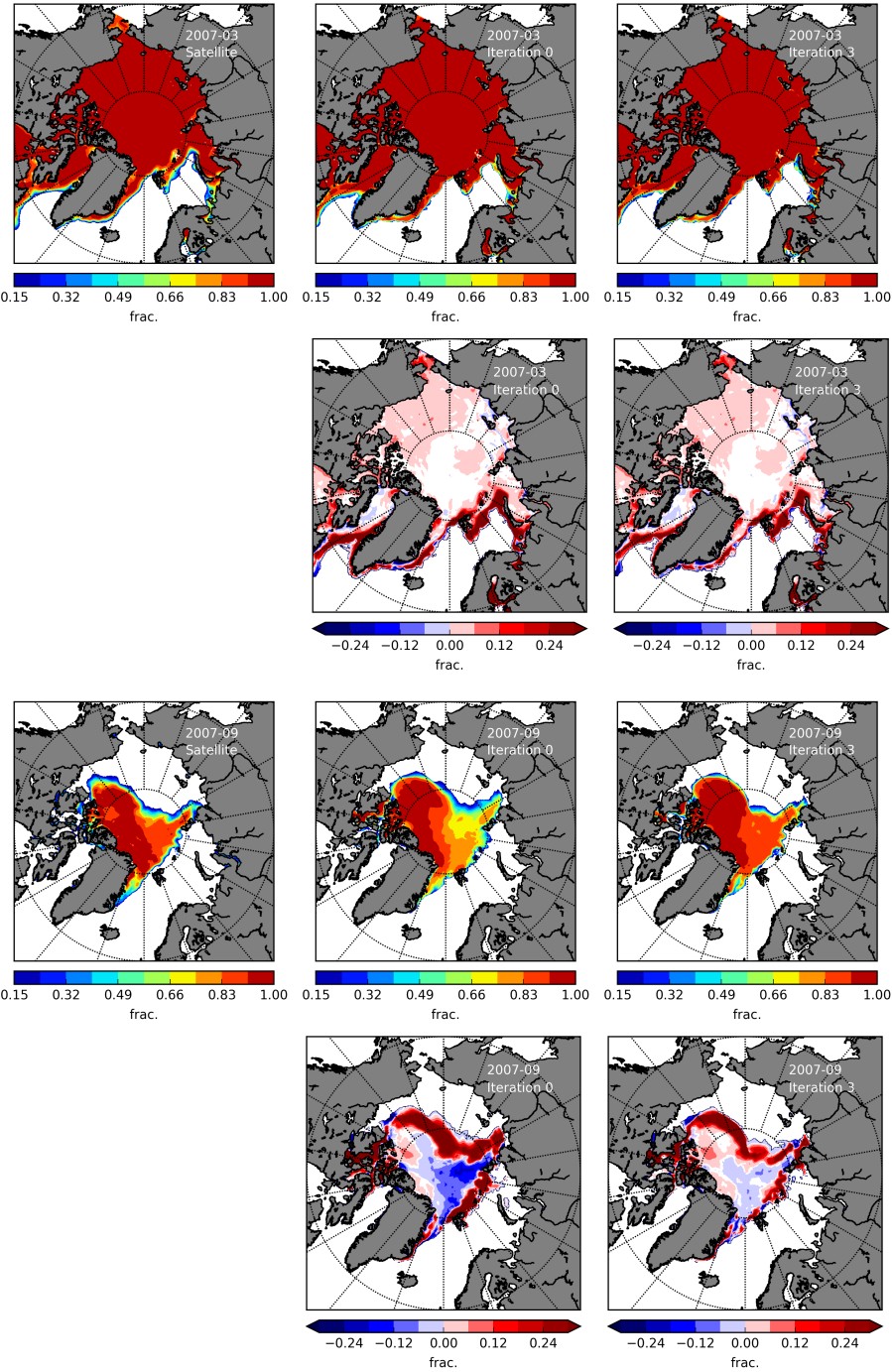

**Figure 4.** Same as Fig. 3, but for year 2007 (the year of the overall minimum sea ice.)

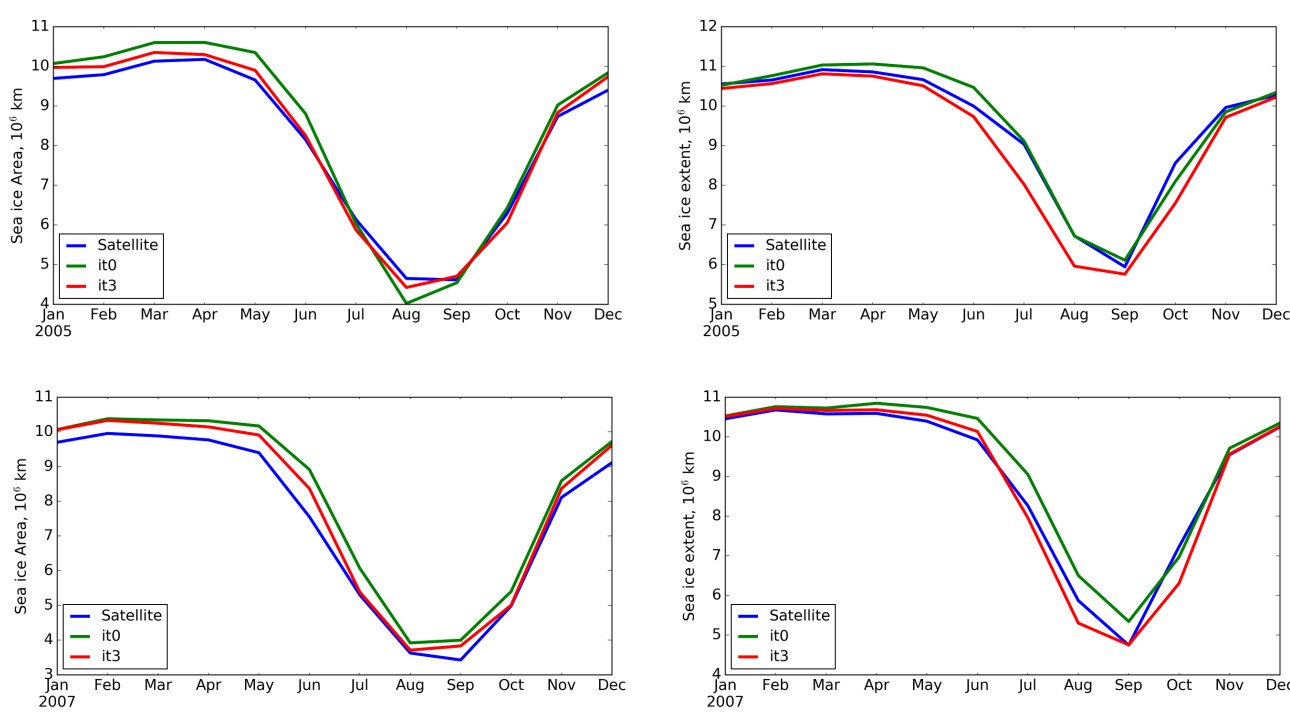

**Figure 5.** Monthly mean sea ice area (left) and extent (right) for the years 2005 (top) and 2007 (bottom). Assimilated satellite data is shown in blue, model solution without corrections is shown in green and the result from the last iteration is shown in red.

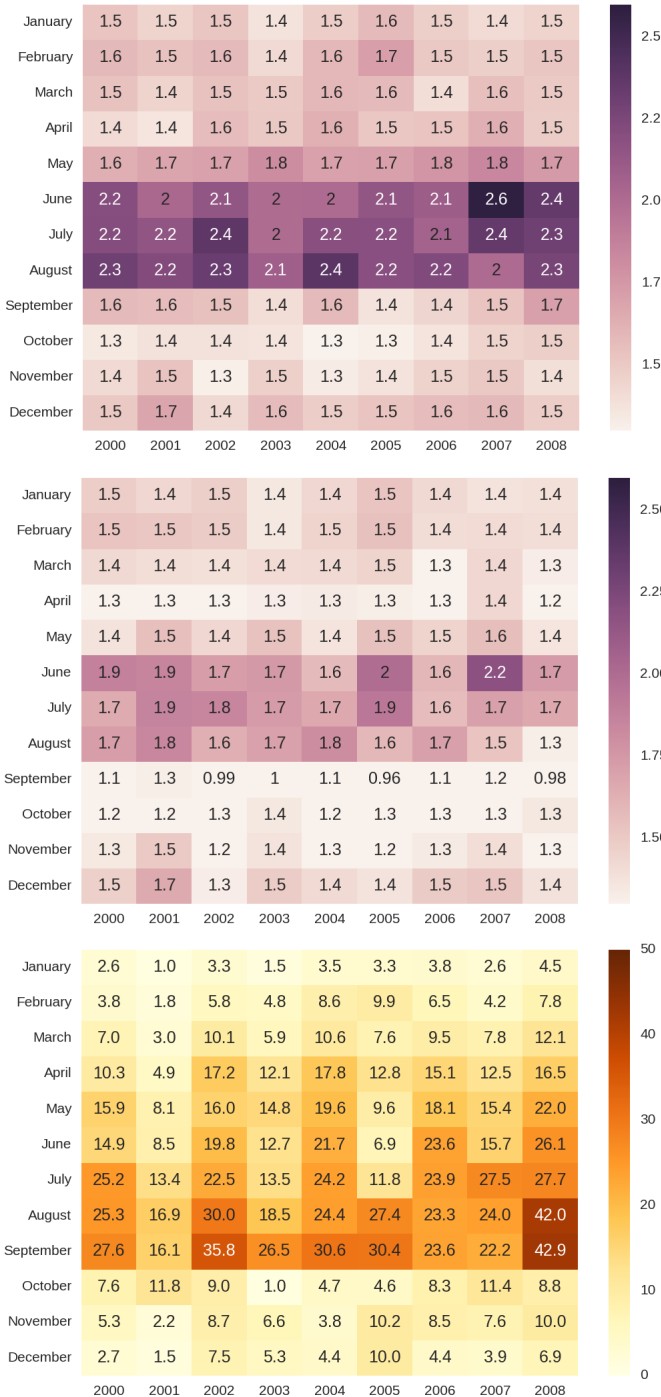

**Figure 6.** Sum of the sea ice area absolute differences (SoAD), compared to assimilated sea ice at every grid location, for every month (in $10^6$ km$^2$), before assimilation (top), after assimilation (middle) and the percent difference between the two (bottom). Positive differences correspond to a decrease of SoAD.

| | 2000 | 2001 | 2002 | 2003 | 2004 | 2005 | 2006 | 2007 | 2008 |
|---|---|---|---|---|---|---|---|---|---|
| January | 1.3 | 1.4 | 1.3 | 1.3 | 1.3 | 1.5 | 1.4 | 1.3 | 1.3 |
| February | 1.4 | 1.5 | 1.4 | 1.4 | 1.5 | 1.6 | 1.5 | 1.4 | 1.4 |
| March | 1.3 | 1.4 | 1.4 | 1.4 | 1.5 | 1.5 | 1.3 | 1.4 | 1.4 |
| April | 1.4 | 1.2 | 1.4 | 1.3 | 1.5 | 1.4 | 1.4 | 1.3 | 1.4 |
| May | 1.3 | 1.3 | 1.4 | 1.4 | 1.5 | 1.3 | 1.5 | 1.4 | 1.4 |
| June | 1.6 | 1.7 | 1.7 | 1.6 | 1.6 | 1.9 | 1.8 | 1.8 | 1.9 |
| July | 2.3 | 2.3 | 2.4 | 2.2 | 2.2 | 2.4 | 2.3 | 3.2 | 2.3 |
| August | 2.2 | 2.2 | 1.9 | 2.1 | 2.1 | 1.9 | 1.9 | 2.5 | 2.4 |
| September | 1.8 | 1.8 | 1.9 | 1.9 | 1.6 | 1.5 | 1.6 | 1.9 | 2.3 |
| October | 1.4 | 1.5 | 1.3 | 1.5 | 1.4 | 1.8 | 1.5 | 1.9 | 1.4 |
| November | 1.5 | 1.5 | 1.6 | 1.5 | 1.6 | 1.7 | 1.5 | 1.6 | 1.4 |
| December | 1.4 | 1.4 | 1.3 | 1.4 | 1.3 | 1.4 | 1.4 | 1.2 | 1.2 |

| | 2000 | 2001 | 2002 | 2003 | 2004 | 2005 | 2006 | 2007 | 2008 |
|---|---|---|---|---|---|---|---|---|---|
| January | 1.3 | 1.4 | 1.3 | 1.3 | 1.3 | 1.5 | 1.4 | 1.3 | 1.3 |
| February | 1.4 | 1.4 | 1.4 | 1.3 | 1.4 | 1.4 | 1.4 | 1.3 | 1.3 |
| March | 1.3 | 1.3 | 1.3 | 1.3 | 1.4 | 1.4 | 1.2 | 1.3 | 1.3 |
| April | 1.2 | 1.2 | 1.2 | 1.3 | 1.2 | 1.2 | 1.2 | 1.2 | 1.2 |
| May | 1.2 | 1.2 | 1.2 | 1.2 | 1.1 | 1.2 | 1.2 | 1.2 | 1.1 |
| June | 1.4 | 1.6 | 1.5 | 1.5 | 1.4 | 2 | 1.4 | 1.6 | 1.7 |
| July | 2 | 2.1 | 2.2 | 2.1 | 1.8 | 2.7 | 1.8 | 2.6 | 2 |
| August | 1.9 | 1.9 | 2 | 1.9 | 1.8 | 1.7 | 1.5 | 1.5 | 1.9 |
| September | 1.3 | 1.5 | 1.3 | 1.3 | 1.1 | 1.1 | 1.2 | 1.3 | 1.2 |
| October | 1.4 | 1.3 | 1.3 | 1.6 | 1.5 | 1.9 | 1.4 | 2 | 1.6 |
| November | 1.4 | 1.5 | 1.5 | 1.4 | 1.5 | 1.5 | 1.4 | 1.5 | 1.3 |
| December | 1.4 | 1.4 | 1.2 | 1.3 | 1.3 | 1.2 | 1.3 | 1.2 | 1.2 |

| | 2000 | 2001 | 2002 | 2003 | 2004 | 2005 | 2006 | 2007 | 2008 |
|---|---|---|---|---|---|---|---|---|---|
| January | 2.0 | 0.3 | 2.7 | 0.4 | 3.3 | 3.1 | 3.6 | 3.2 | 2.2 |
| February | 2.3 | 1.2 | 4.3 | 0.9 | 8.7 | 11.4 | 5.7 | 4.3 | 5.3 |
| March | 1.1 | 3.1 | 3.4 | 1.2 | 5.7 | 6.6 | 5.5 | 6.2 | 7.7 |
| April | 9.3 | 1.6 | 13.6 | 2.1 | 16.6 | 14.3 | 12.2 | 11.4 | 13.3 |
| May | 10.1 | 8.1 | 16.2 | 13.2 | 22.6 | 8.7 | 15.6 | 12.8 | 16.2 |
| June | 12.0 | 6.8 | 9.3 | 6.3 | 14.7 | -5.4 | 19.5 | 14.2 | 7.8 |
| July | 12.4 | 9.6 | 7.8 | 6.4 | 17.4 | -9.5 | 24.2 | 18.0 | 11.3 |
| August | 14.7 | 16.1 | -5.0 | 9.5 | 15.9 | 11.3 | 19.1 | 38.1 | 21.5 |
| September | 27.4 | 17.4 | 33.8 | 31.8 | 29.4 | 25.5 | 25.9 | 30.4 | 47.9 |
| October | 1.7 | 12.5 | 0.2 | -10.3 | -1.1 | -9.3 | 2.8 | -3.0 | -9.0 |
| November | 3.2 | 1.9 | 4.7 | 3.0 | 2.8 | 9.1 | 6.4 | 6.8 | 1.8 |
| December | 0.9 | 1.1 | 7.7 | 4.2 | 3.4 | 12.0 | 2.7 | 2.4 | 3.9 |

**Figure 7.** Same as Fig. 6, but for the sea ice extent.

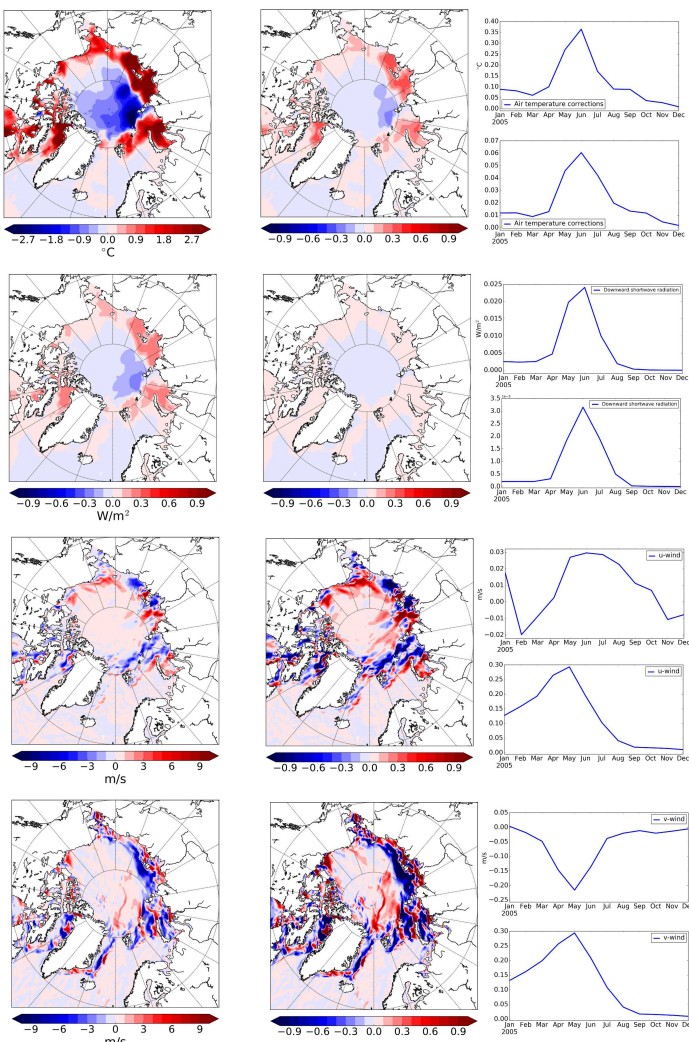

**Figure 8.** Corrections for different surface forcing variables: spatial distribution (left) and spatial distribution scaled by the uncertainty (middle) for the month with the largest absolute value of corrections in 2005. Also shown is the monthly climatology for the sea ice area mean corrections (right column) averaged over the area north of 66.5° N (top panels for each variable) and the average of absolute values scaled by the uncertainties (lower panel for each variable). Corrections are shown for June 2005 2-m air temperature (first row), June 2005 downward shortwave radiation (second row), June 2005 zonal component of the wind (third row) and May 2005 meridional component of the wind (fourth row).

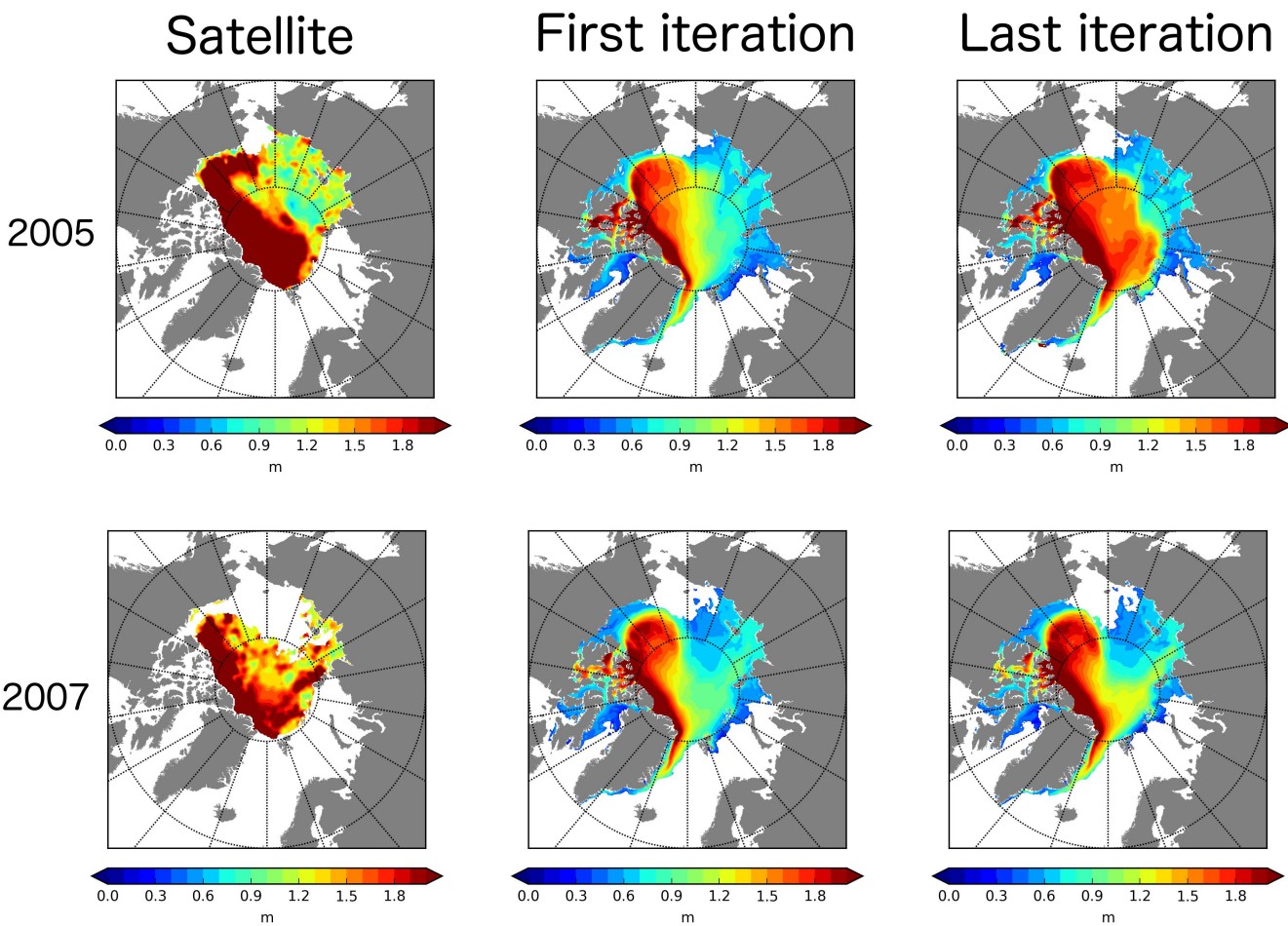

**Figure 9.** Sea ice thickness in October-November for years 2005 (top row) and 2007 (bottom row). Left column presents satellite data (ICESat, Kwok data); middle column are model results before assimilation (first iteration); right column corresponds to model results after assimilation (last iteration).

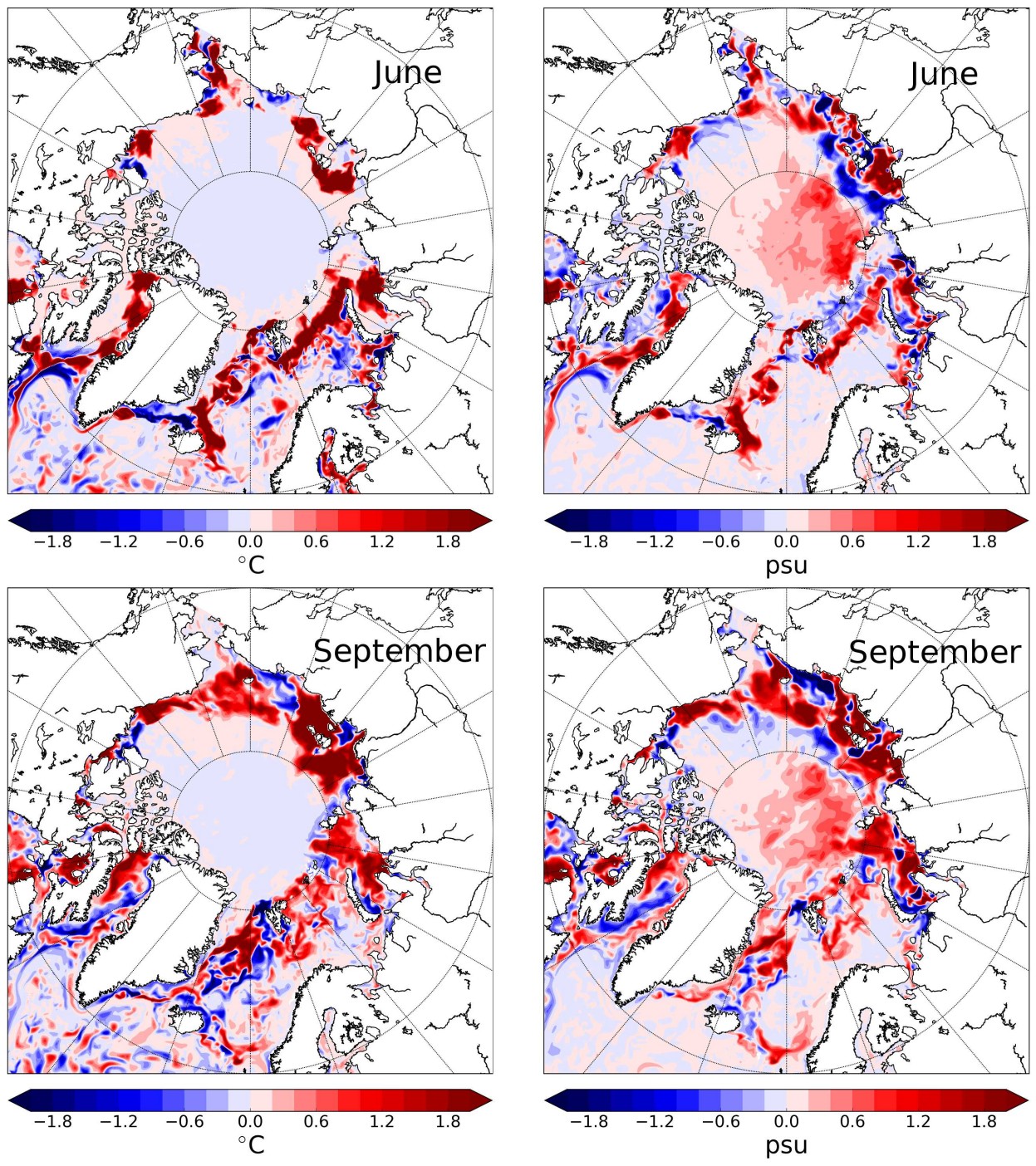

**Figure 10.** Differences in ocean surface temperature (left column) and salinity (right column) between first guess and last iteration for June 2005 (top row) and September 2005 (bottom row).

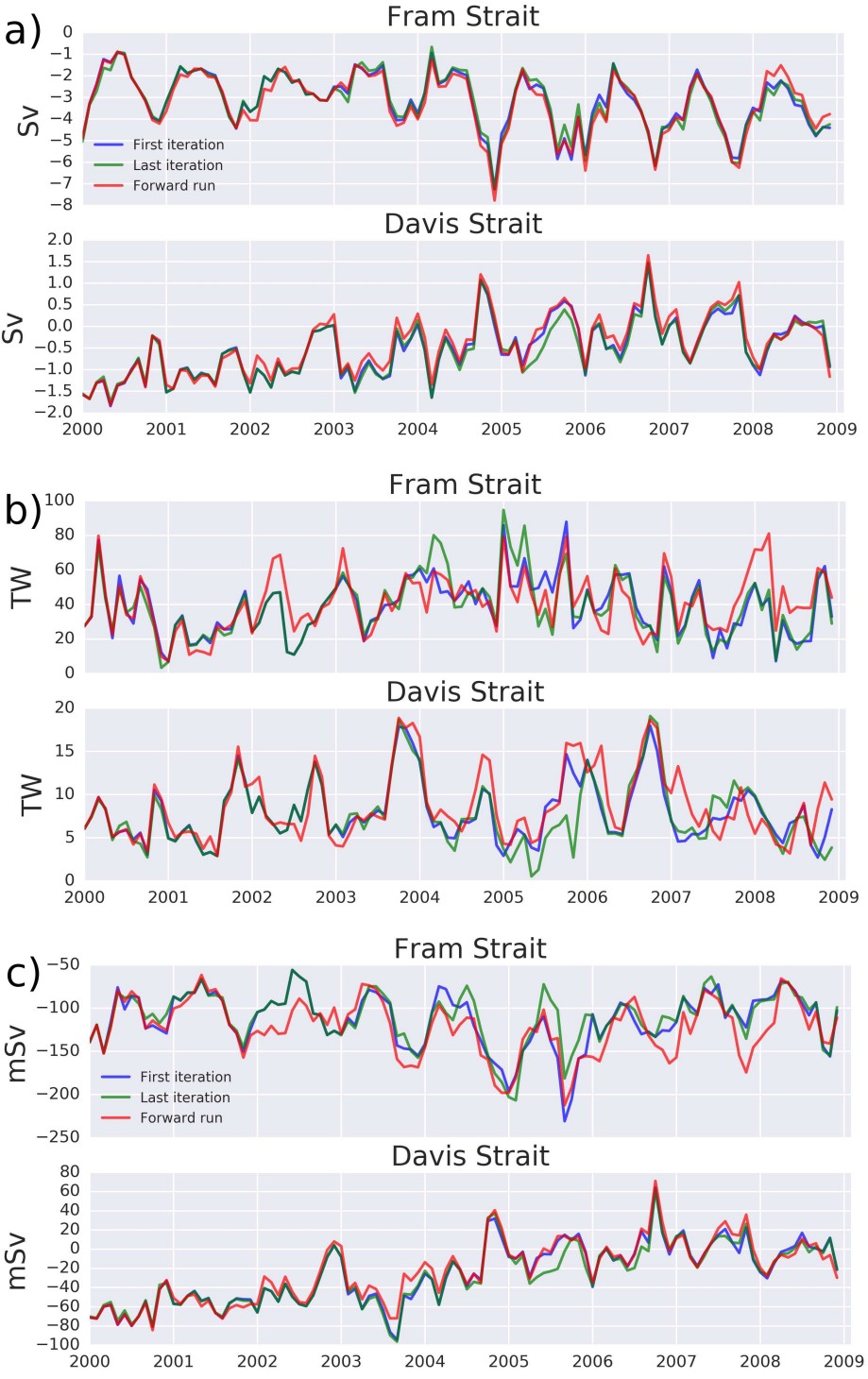

**Figure 11.** Fluxes through the Fram and Davis Straits of (a) volume, (b) heat and (c) freshwater. Positive fluxes are into the Arctic Ocean. Results are shown for the forward run (red), for the run before assimilation (blue) and for the run after assimilation (green).