# Peer review of "Sea Ice Assimilation into a Coupled Ocean-Sea Ice Model Using its Adjoint"

_The Cryosphere, 2017_

## Referee Comment (RC1) · Anonymous Referee #1 · 23 Feb 2017

Review comments on "Sea Ice Assimilation into a Coupled Ocean-Sea Ice Adjoint Model of the Arctic Ocean" by Koldunov et al.

1. Summary

The paper deals with an ocean-sea ice data assimilation experiment in the Arctic Ocean by an adjoint method. The data used for the assimilation are ocean hydrography (in-situ and remotely sensed measurements and climatology) and sea ice concentration. These data are assimilated to a regional coupled ocean-sea ice model (MITgcm + Hibler-type sea ice model) for a state estimation of 2000 - 2008 period, by adjusting the initial and boundary conditions for the model (the control vector is composed of ocean initial conditions and atmospheric boundary conditions). The authors report substantial improvement of modeled sea ice concentration and position of ice edge in summer, as

well as improvement of seasonal cycle of sea ice cover and ice thickness distribution. The authors also report that the improvements occur mainly due to corrections to the surface atmospheric temperature.

**2. General comments**

The adjoint method is one of the promising data assimilation methodologies for state estimations, since the method can preserve modeled physics in the estimated state. The estimated state is not only usable as a 4-dimensional interpolation of observation, but also applicable for dynamical interpretations of the system. However, setting up the adjoint model, which has the consistent physics with the corresponding forward model, sometimes requires substantial efforts, and in addition running the adjoint model with a long assimilation window needs linearization of the code. The authors seem to suffer from these technical issues, although the details were not provided in the manuscript. They run the adjoint model without some ocean modules nor sea ice dynamics. The latter issue (adjoint run without sea ice dynamics) is crucial for the current experiment design, since the study focuses on improvement of sea ice status and associated corrections to the control variables. In addition, the authors divide the 2000 - 2008 assimilation window into 1 year chunk. This division seriously deteriorates the advantage of the adjoint method, since the observed data cannot contribute to improve the model status of preceding years. This issue may affect one of the conclusions of this study that the authors don't find sizable improvements of ocean field, since the spin-up/spin-down time for ocean is much longer than that for sea ice. For these reasons, the present study did not take the advantage of the adjoint method. Since the authors did not provide the details of the experiment design (e.g., definition of the cost function, penalty term, uncertainty for each observational data), some of the results in the manuscript are difficult to interpret. As a whole, unfortunately I cannot find any new technical achievement nor sound scientific findings for Arctic ocean-sea ice system in this manuscript.

**3. Major points**
- Why the authors switched off the sea ice dynamics of the adjoint model? Since the main focus of this study is to examine the effect of sea ice concentration assimilation on the modeled field and correction to the control variables, the adjoint run without sea ice dynamics may seriously deteriorate the results. Particularly, I have concerns about the result that the system obtained the optimal solution of sea ice concentration by the corrections to the 2 m air temperature. Since the adjoint model doesn't take the sea ice dynamics into account, the dynamical forcing, such as wind forcing and/or ocean drag, cannot directly contribute to improve sea ice concentration by the assimilation. I would guess this is one of the reasons why the ice concentration improvement occurs mainly due to thermal forcing (2 m air temperature), and occurs not by dynamical forcing. To obtain the conclusions in this study, the sea ice dynamics is indispensable in the adjoint model.

- The division of the 2000 - 2008 assimilation window into 1 year chunks deteriorates the advantage of the adjoint method. Due to the current set-up of the system, the system cannot use the observed data to improve the model status for preceding years. I am afraid that this short assimilation window might be a cause of the small improvement of ocean status compared to sea ice in the current assimilation (ocean needs longer spin-up/spin-down time than sea ice, and 1 year window is too short even for layers shallower than Arctic halocline). I also would like to point out that even for sea ice, 1 year chunk is too short, if the authors intend to examine improvement of ice thickness. In order to extend the assimilation window, I think further linearization of the code (including consistency check with the original code) is necessary.

- The authors should describe how the cost function is defined (i.e., the objective function, the gradient of which is estimated by the adjoint), since the definition of the cost (i.e., weighting between different types of observed data, error estimates for observational data, and definition of penalty term, etc.) strongly affect the behavior of the system. Due to the lack of these information, some results shown in the manuscript are difficult to interpret.

[Figure]

- How do the authors control or constrain the allowable corrections to the control variables? Since the spatial pattern of the correction to the 2 m air temperature (Fig. 8) is quite similar to the bias of the free run (Fig. 3), I have a concern that the system changes the control variables without reasonable constraint. The authors cite Köhl (2015) as a reference for the atmospheric condition. In this paper, the errors of atmospheric field are prescribed by the standard deviation of the NCEP field. What does this mean? The standard deviation of the NCEP field is the ensemble spread of the NCEP climate model? If so, the standard deviation does not provide error of the reanalysis field, but provides the magnitude of the natural variability of the modeled climate system. Since the adjoint method tries to impose all the deficiencies of the model status to the control variables, the validity of the allowable corrections to the control variables should be carefully examined.

- What is the new technical achievement or new scientific finding(s) of this study, particularly, in comparison with the number of former sea ice (and partly ocean) data assimilation studies for Arctic region? Since the sea ice dynamics of the adjoint model is not consistent with the forward model and the assimilation window is only 1 year, I hardly find any advantages on this experiment compared to the former studies using another assimilation method (e.g., optimal interpolation, 3D-Var, Green function method and EnKF).

4. Minor points

- Page 4, line 16: Which module is excluded from the adjoint code? What effect do the authors expect by this exclusion, particularly, in relation to the conclusion of this study?

- Page 4, line 16-17: Why the sea ice dynamics are switched off? Due to the absence of sea ice dynamics, the adjoint model cannot provide the optimal gradient, and therefore the correction to the control variable may not reflect (modeled) reality.

- Page 4, line 20-23: Are the control variables used to define penalty term in the cost (object) function? Please provide the exact definition of the penalty term for reproducibility of the study.

- Page 4: Please describe exact form of the cost function (including penalty terms) used in this study.

- Page 4, line 23: How did the authors define the uncertainty for hydrographic (e.g., EN3, NISE, etc.) and satellite data? Although the author cited Köhl (2015) as a reference, the representation error of in-situ measurements should be different since the resolution of the model differs.

- Page 4, line 24: Why did the authors apply constant error of 50% for sea ice concentration? OSI-SAF provides uncertainty estimates of ice concentration at every grid points. The constant error does not take into account the large uncertainties over the marginal ice zone, while it underestimates weight of reliable data over the central arctic.

- Page 4, line 24: Why the data assimilation is performed in one year chunk? Due to this experiment design, the authors cannot take the advantage of adjoint method. Particularly, the authors cannot examine the effect of data assimilation to the ocean variables, since 1 year chunk is too short even for layers above Arctic halocline.

- Page 5, line 8-10: How did the authors define the relative weight between different types of observation? Satellite measurements generally cover the large area with constant time interval, whereas the errors of the data are not independent but covariant. On the other hand, in-situ ocean measurements are very sparse, while the measurement errors are almost negligible (and therefore assumed to be independent each other) compared to the representation error. In addition, the magnitude of the representation error depends on the size of model's grid cell. How did the authors handle these issues? Description is needed.

- Page 6, line 23-25: I don't understand the meaning of this sentence. More explanations are needed.

- Page 6, line 28-30: Please explain relation between the Hausdorff distance (Dukohvskoy et al. 2015) and the metrics used in this study (sum of the RMS errors). I do not understand why the authors introduced Dukohvskoy et al., (2015) here.

- Page 7, line 18-24: It is hard to believe that the spatial pattern of the bias of 2 m air temperature of NCEP reanalysis coincide with that of modeled ice concentration bias. If I understand correctly, the uncertainties of the NCEP reanalysis data used in this study are given by the standard deviation of ensemble runs of NCEP climate model. Does the difference between the ensembles has such a sharp gradient like the correction to the 2 m air temperature (Fig. 8 the first row, left)?

- Page 7-8: The authors described that the correction to the 2 m air temperature is the main driver to improve the sea ice concentration in the assimilated field, and the contributions from other control variables, such as wind forcing, are very small. I am afraid this may be an artifact due to the lack of sea ice dynamics in the adjoint model, as described in major point.

- Page 7, line 31-33: The wind can play a role not only in local redistribution of the sea ice along the shore and ice edge, but also in large-scale sea ice distributions, although such effect is not seeable in the present experiment design.

- Page 8, line 8: ".., both making atmospheric forcing actually worse". I do not understand the meaning of this sentence.

- Page 9, line 24-25: This is interesting. Could the authors provide specifications of the mechanism?

- Page 9, line 33 - Page 10, line 1: How much is the ratio of relative contributions to the cost function between ocean variables and sea ice variable? If the contribution from sea ice variables dominates, the system tries to change the control variables which have large impact on sea ice, and then it is natural that the changes of the ocean variables are small.

- Page 10, line 1-3: The fluxes shown in Fig. 11 are improved by the assimilation?

As far as I know there are some flux estimates through Arctic gateways based on observations (e.g., Tsubouchi et al., 2014, JGR).

- Page 10, line 25-28: see the comments above.

- Page 11, line 3-4: This result is interesting, but I am still afraid that this improvement might be achieved by wrong reason, due to the absence of sea ice dynamics in the adjoint, since the ice dynamics is important for the redistribution and accumulation of sea ice.

- Page 11, line 11-13: I agree that the estimated state of sea ice is consistent with the modeled physics, whereas due to the lack of ice dynamics in the adjoint, we cannot make sure the correction to the control variables are realistic. In other words, we cannot exclude a possibility that the estimated state is achieved by artificial forcing different from reality, and therefore by thermodynamic and dynamical balance different from reality.

- Figure 3 caption, line 3: "third" should be "forth".

- Figure 3, 4, 8 and 10: It would be helpful for comparison of the spatial patterns, if the longitude and latitude lines (as in Fig. 9) are embedded in these figures.

---

## Referee Comment (RC2) · Anonymous Referee #2 · 13 Mar 2017

Summary

In this work the authors demonstrate the synthesis of hydrographic and sea ice concentration data into a 16-km horizontal resolution Arctic and North Atlantic coupled sea ice-ocean model. The reduction of an uncertainty-weighted model-data difference cost function was achieved by iteratively optimizing a set of adjustments to a set of atmospheric and initial condition control variables using gradient information provided by the adjoint of the numerical sea ice-ocean model. The final multiyear state estimate was constructed by optimizing each single year between 2000 and 2008 in succession - the final optimized state of year X is defines the initial state for year X+1.

The authors demonstrate improvements of the model's reproduction of the data. The largest reduction in terms of percentage is found with sea ice concentration and SST

with lower relative cost reduction for other data, including T and S profiles, SSH, and mean dynamic topography. The largest sea ice concentration cost reductions in terms of RMS are found during summer months. Discrepancies between simulated and observed sea ice extent are found to increase in some months even when discrepancies in simulated and observed total sea ice area decrease. After synthesizing ocean and sea ice data, little impact is seen in ocean volume, heat, and freshwater fluxes through Fram Strait and Davis Strait.

Specific Comments

1) With respect to the title, assimilation is not "into a Coupled Ocean-Sea Ice Adjoint Model". The assimilation is "into a Coupled Ocean-Sea Ice Model using its adjoint".

2) Abstract: Better to provide the actual spatial resolution of the satellite sea ice concentration data that is assimilated rather than refer to it as 'high resolution'.

3) Page 1, Line 6: 'values of sea ice extent become underestimated' doesn't define a metric. Is the metric the sum of model minus data or weighted model minus data difference or the RMS of model minus data or something else?

4) Page 1, Line 6-7: Characterizing a state estimate of a system as complex as the Arctic Ocean requires that one analyzes a suite of metrics. The author's statement that one the sea ice extent metric is "not suitable to characterize the quality of the sea ice simulation" is odd and out of place. To whom is this statement aimed? This seems to be a straw man argument.

5) Page 1, Lines 10-11: The atmospheric control variable adjustments that one finds during any optimization are intimately related to the magnitudes of the prior uncertainties of the individual terms of the first-guess atmospheric state. The author's statement that biases in sea ice are reduced 'mainly due to corrections to the surface atmosphere temperature' is difficult to interpret because the reader does not know the magnitude of prior uncertainties used during e optimization. Are you referring to the sum of the

squared normalized adjustments? How is surface atmosphere temperature identified as the main control variable correction since atmospheric forcing units are arbitrary?

6) Page 2, Line 8. The authors may consider using the term 'state estimation' to describe the model-data synthesis methodology used in this study instead of the term 'data assimilation'. An uninformed reader may think that the work conducted here referring to sequential data assimilation, a technique that has been applied to sea ice data for decades. The adjoint method used in this work is rather special and yields quite a different product (namely a physically-consistent ocean and sea ice state). Below I post an excerpt from Wunsch and Heimbach, 2007 in which they argue for their choice of the term 'state estimation' when describing the application of the adjoint method to combine data with a model (emphasis mine):

"In physical oceanography, the problem of combining observations with numerical models differs in a number of significant ways from its practice in the atmospheric sciences. It is these differences that lead us to use the terminology "state estimation" to distinguish the oceanographers' problems and methods from those employed under the label "data assimilation" in numerical weather prediction. "Data assimilation" is an apt term, and were it not for its prior use in the meteorological forecast community, it would be the terminology of choice. But meteorologists, faced with the goal of daily weather forecasting, have developed sophisticated techniques directed at their own particular problems, along with an opaque terminology not easily penetrable by outsiders. Because much of oceanography has goals distinct from forecasting, the direct application of meteorological methods is often not appropriate."

7) Page 4 Line 10-11: List the control variables.

8) P4 Line 20: Describe why the atmospheric control variable frequency was changed to daily.

9) P4 Line 23: As atmospheric adjustments are an important control parameter in this work, the authors should (a) explicitly state how they were derived as Kohl (2015): "For

the atmospheric state, errors are calculated as before from the [standard deviation] of the NCEP fields." And (b) show maps of their magnitudes in the main text or in supplemental materials. Also, because they are so important, more discussion about your choice of standard deviation of NCEP fields is appropriate. The standard deviation of Arctic near-surface atmosphere temperatures is considerable given the large seasonal cycle. In much earlier versions of ECCO/GECCO the use of atmospheric state standard deviations could be justified because in mid-latitudes and the tropics they partially captured "random" variations due to synoptic variability. At high latitudes the standard deviation for near-surface atmosphere temperature and shortwave radiation is mostly due to the seasonal cycle.

10) Page 4, Line 24: Why are the sea ice data assigned a constant 50% error? Satellite SIC products have errors that are far smaller than that everywhere except in the MIZ and in summer when meltponds are present.

11) Page 4, Line 27: To clarify, each year after the first uses initial conditions that are identical to the final state of the previous year, correct?

12) Page 4, Lines 3-4: Some SST products have nonzero values beneath sea ice. Is that the case in the RSS dataset?

13) Table 1: I understand that the PHC climatology had large biases relative to modern Arctic T and S because it was derived with observations mainly from the 1970's and 1980's and before the recent shifts in Arctic heat and freshwater (McPhee et al, 2009). Can you comment on how the simultaneous use of the PHC climatology alongside contemporary data may have affected the T and S cost reduction?

14) Page 4, Paragraph 1: Cost function reduction percentages are important but obviously they are dependent on how close to the data you were when you began your simulations. The first-guess solution of Fenty et al., (2015) could have been further from the data than your first-guess solution. While both may end up in the same state, their reduction percentage would be higher. The most important information is how

well one's final state estimate fits the data. Much less important is the magnitude of the improvement relative to one's (somewhat arbitrary) starting point.

15) Page 4, before line 31: It may be useful to mention how many iterations were conducted before the 1% threshold was achieved. In Figure 3 I see "iteration 3" as the final iteration for 2005 and 2007. That strikes me as unusual. If your cost was dominated by SIC and SST data and the adjoint method quickly reduced the misfits of those data, then I can see how you hit the 1% total cost reduction threshold quickly. However, it is possible that if those two datasets were ignored, the adjoint machinery could have continued to substantially reduce misfits in other datasets. Can you comment on that?

16) Page 5, Line 15: There may be a missing figure. I cannot match up Figure 2 to the description offered here. Fig 2 is % cost reduction in different years vs. data.

17) Page 5, Line 24: Good to additionally mention why most models overestimate sea ice in the Greenland Sea with a reference.

18) Page 5, Line 35: This is probably because in these extreme months the location of the sea ice edge is relatively stable compared to spring and fall months when the ice pack contracting and expanding.

19) Section 4: This entire discussion must be rewritten. Atmospheric control variable adjustments seem to be compared by their relative magnitudes but their relative magnitudes are not meaningful because these physical variables have different, arbitrary, units. By all means show the magnitude of the adjustments but to make a meaningful comparison one should first normalize them by their prior uncertainties. a. This includes Figure 8, which should be updated to show all control variable adjustments normalized by their uncertainties. Also include longwave radiation.

20) Page 8, Line 20-22. The "probably realistic" spatial distribution of the Kwok Arctic sea ice thickness field deserves a reference. Are the 0.7 m errors spatially correlated or uncorrelated?

21) Page 9, Line 22-24: Neither the length of the simulation nor the number of T/S profiles is a fundamental impediment to magnitude of model-data misfit reduction. An iteration 0 state with T and S close to the data as measured by the prior uncertainty could be responsible. Maybe averaged normalized costs should be added to Figure 2 for each cost category for iteration 0 and the final iteration.

22) I may be incorrect but it seems that no Arctic Ocean T and S profiles were used in this work. I do not see Arctic Ocean data in the Ingleby and Huddleston report and the NISE database doesn't show data north of the Norwegian Sea. Given that the assimilation period overlaps with the existence of ice-tethered profilers, why were ice-tethered profile data not included (http://www.whoi.edu/page.do?pid=20781)? As for the CTD data in the Arctic, both the ICES database (http://www.ices.dk/marine-data/data-portals/Pages/ocean.aspx) and the World Ocean Database v3 (https://www.nodc.noaa.gov/OC5/WOD13/ ) have data for the time period considered in this work. There may be perfectly fine reasons for excluding these data but the reasons should be offered.

Technical Corrections

1. Page 1, Line 5: change 'become' to 'are' as in 'values of sea ice extent are underestimated' 2. Page 1, Line 5: first comma to semicolon. Or split this long sentence into two before 'however' 3. Page 1, Line 14: strike 'to date' 4. Page 1, Line 16: reference? 5. Page 1, Line 17: strike comma before 'is therefore of utmost importance' 6. Page 1, Line 24: strike 'if not possible' 7. Page 2, line 2, strike comma before 'the community'. Strike 'heavily'. 8. Your doi for Detlef's 2016 paper is incorrect. It should be DOI: 10.1146/annurev-marine-122414-034113 9. Page 2, Line 19: strike "usually in general" 10. Page 5, Line 12: strike "are going to" 11. Page 5, Line 28: replace "very good" with "improved" 12. Page 5, Line 29: strike "thus" 13. Page 5, Line 31-32: This sentence deserves a rewrite for clarity. As mentioned above, relative percentage sea ice cost reductions are also a function of the (unknown) first guess states. 14. Add 'bears' before 'a good resemblance' 15. Page 5, Line 24: For clarity consider saying

'since a perfect total sea ice area evolution...' and the following sentence is redundant. 16. Page 8, Line 20-22. Strike "except for the" and simply say that "Sea ice thickness are not provided by Kwok for the Barents and Kara Seas and the Canadian Archipelago because ..." with a reference. 17. Page 8, Line 26: change "variables" to "variables'" 18. Page 9, Lines 1-2: Why is it hard to provide quantitative estimates? You could plot time series of the uncertainty-weighted squared model-data misfit (normalized cost) before and after the assimilation. 19. Plotting model minus data or model minus data squared in Fig 5 might simplify comparison. 20. Section 4: Fonts on the time series of Fig 8 are also small and difficult to read. One subplot is cut off. After normalizing the summed control variable adjustments they could all be shown in the together in the same plot.

Wunsch, C., & Heimbach, P. (2007). Practical global oceanic state estimation, 230(1–2), 197-208. http://doi.org/10.1016/j.physd.2006.09.040

Fenty, I., & Heimbach, P. (2013). Coupled Sea Ice-Ocean-State Estimation in the Labrador Sea and Baffin Bay. Journal of Physical Oceanography, 43(5), 884-904. http://doi.org/10.1175/JPO-D-12-065.1

McPhee, M. G., Proshutinsky, A., Morison, J. H., Steele, M., & Alkire, M. B. (2009). Rapid change in freshwater content of the Arctic Ocean. Geophysical Research Letters, 36(10). https://doi.org/10.1029/2009GL037525

---

## Author Comment (AC1) · 5 Jun 2017

**Reviewer 1**

Review comments on "Sea Ice Assimilation into a Coupled Ocean-Sea Ice Adjoint Model of the Arctic Ocean" by Koldunov et al.

**1. Summary**

*The paper deals with an ocean-sea ice data assimilation experiment in the Arctic Ocean by an adjoint method. The data used for the assimilation are ocean hydrography (in-situ and remotely sensed measurements and climatology) and sea ice concentration. These data are assimilated to a regional coupled ocean-sea ice model (MITgcm + Hibler-type sea ice model) for a state estimation of 2000 - 2008 period, by adjusting the initial and boundary conditions for the model (the control vector is composed of ocean initial conditions and atmospheric boundary conditions). The authors report substantial improvement of modeled sea ice concentration and position of ice edge in summer, as well as improvement of seasonal cycle of sea ice cover and ice thickness distribution. The authors also report that the improvements occur mainly due to corrections to the surface atmospheric temperature.*

We thank the Reviewer for thoroughly evaluating our manuscript. In the following Reviewer's comments are in italic, our answers are in usual font, text from the manuscript is in the quotation marks and the new text is in blue color.

**2. General comments**

*The adjoint method is one of the promising data assimilation methodologies for state estimations, since the method can preserve modeled physics in the estimated state. The estimated state is not only usable as a 4-dimensional interpolation of observation, but also applicable for dynamical interpretations of the system. However, setting up the adjoint model, which has the consistent physics with the corresponding forward model, sometimes requires substantial efforts, and in addition running the adjoint model with a long assimilation window needs linearization of the code. The authors seem to suffer from these technical issues, although the details were not provided in the manuscript. They run the adjoint model without some ocean modules nor sea ice dynamics. The latter issue (adjoint run without sea ice dynamics) is crucial for the current experiment design, since the study focuses on improvement of sea ice status and associated corrections to the control variables. In addition, the authors divide the 2000 - 2008 assimilation window into 1 year chunk. This division seriously deteriorates the advantage of the adjoint method, since the observed data cannot contribute to improve the model status of preceding years. This issue may affect one of the conclusions of this study that the authors don't find sizable improvements of ocean field, since the spin-up/spin-down time for ocean is much longer than that for sea ice. For these reasons, the present study did not take the advantage of the adjoint method. Since the authors did not provide the details of the experiment design (e.g., definition of the cost function, penalty term, uncertainty for each observational data), some of the results in the manuscript are difficult to interpret. As a whole, unfortunately I cannot find any new technical achievement nor sound scientific findings for Arctic ocean-sea ice system in this manuscript.*

We agree with the Reviewer that not all of the possible advantages of the adjoint assimilation method were used in this study and that by employing all of them the study would benefit greatly. However, as the Reviewer have mentioned, getting the adjoint assimilation to work is a technically challenging task and it is hard to expect that all the technical issues would be solved in the pilot study we present here, in which we are only beginning to gain experience with adjoint sea ice assimilation.

Saying that the present study "*did not take the advantage of the adjoint method*", while we agree in principle that long-term processes are not present in the sensitivities, at the same time is in our opinion an overstatement. Adjoint assimilation techniques are successfully used in ocean and atmospheric sciences with short assimilation windows and certain approximations of the full forward model are made in the adjoint mode.

It seems that a lot of the Reviewer's judgment about our results is based on the erroneous assumption that the sea ice in the adjoint model was not allowed to move. We admittedly made a bad selection of words by saying in the "Methods" section that the "sea ice dynamics were switched off" in the adjoint mode. What should have been said is that the sea ice is in the "free drift" mode and that the ice rheology is not taken into account.

It certainly is wishful to have rheology included as well as being able to benefit from long assimilation windows. However, the adjoint method has strong limitations for nonlinear systems, which cannot easily be circumvented and that unfortunately prevent taking full advantage of capabilities one might expect from applications with more linear models. In particular, the initial goal was to do the assimilation in one sweep and we tried extending the window to periods longer than one year. Unfortunately, the gradient information was no longer useful for the improvement of the state. All this explanation is now included in the text. In addition, following the Reviewer's request, we also added more information about the design of our experiment.

More detailed answers to reviewer's criticism are provided below.

**3. Major points**
- *Why the authors switched off the sea ice dynamics of the adjoint model? Since the main focus of this study is to examine the effect of sea ice concentration assimilation on the modeled field and correction to the control variables, the adjoint run without sea ice dynamics may seriously deteriorate the results. Particularly, I have concerns about the result that the system obtained the optimal solution of sea ice concentration by the corrections to the 2 m air temperature. Since the adjoint model doesn't take the sea ice dynamics into account, the dynamical forcing, such as wind forcing and/or ocean drag, cannot directly contribute to improve sea ice concentration by the assimilation. I would guess this is one of the reasons why the ice concentration improvement occurs mainly due to thermal forcing (2 m air temperature), and occurs not by dynamical forcing. To obtain the conclusions in this study, the sea ice dynamics is indispensable in the adjoint model.*

As stated above, we have only switched off the sea ice rheology, so that the sea ice is in a free drift mode. We admittedly did a very poor selection of words by saying that "sea ice dynamics were switched off" and it is fixed now in the text as follows:

"The sea ice module was active in the adjoint integration, but the part of the sea ice dynamics that treats rheology was switched off, so the sea ice is in a free drift configuration."

The reason for switching off rheology was that we were not able to get useful sensitivities for the sea concentration for long periods of time with the rheology on. This is undoubtedly a flaw in our methodology, but simplification of the adjoint model to provide more useful gradients is a common practice in the adjoint data assimilation community. We note that in the published study of Fenty et al. (2015) similar simplifications were done to the adjoint model. Liu et al. (2012) showed that removing a certain part from the adjoint has little effect

on the adjoint model during short time periods, while it prevents the adjoint integration to become useless on longer time scales.

Realizing that in a free drift mode the model does not consider internal sea ice stresses, the dynamical forcing actually seems to contribute more to the improvements of the sea ice than if the rheology would have been switched on. This actually emphasizes corrections to the wind stress in comparison to other components. As a result of the additional analysis of the corrections to the controls (requested by Reviewer 2), we have removed the statement about the relative contribution of thermo-dynamical forcing to the improvement of the model state.

*- The division of the 2000 - 2008 assimilation window into 1 year chunks deteriorates the advantage of the adjoint method. Due to the current set-up of the system, the system cannot use the observed data to improve the model status for preceding years. I am afraid that this short assimilation window might be a cause of the small improvement of ocean status compared to sea ice in the current assimilation (ocean needs longer spin-up/spin-down time than sea ice, and 1 year window is too short even for layers shallower than Arctic halocline). I also would like to point out that even for sea ice, 1 year chunk is too short, if the authors intend to examine improvement of ice thickness. In order to extend the assimilation window, I think further linearization of the code (including consistency check with the original code) is necessary.*

The reasons for choosing 1 year-long chunks are purely technical in nature (this will be explained in more detail in the answer to a specific question below). The limited time scale of the assimilation is a valid concern, however, as it is shown in our work and in Fenty et al. (2015), one year is enough to significantly reduce model-data differences for the sea ice and not only in one specific year (as in Fenty et al., 2015), but consistently for several years in the 2000s.

The small assimilation window certainly limits the extent to which the data can affect the state. Particularly, processes that act on long time scales such as in the interior of the ocean will not adjust much, which may explain the small ocean improvements. We believe that more important is the much larger amount of sea ice concentration and SST data compared to the very limited amount of data in the Arctic Ocean water column, although one may also hope for ocean improvements though the assimilation of sea ice data. Reviewer 2 proposed a similar explanation below. This conclusion is based on the spatial distribution of adjustments to control variables that obviously mostly reduce discrepancies between modelled and observed sea ice area.

*- The authors should describe how the cost function is defined (i.e., the objective function, the gradient of which is estimated by the adjoint), since the definition of the cost (i.e., weighting between different types of observed data, error estimates for observational data, and definition of penalty term, etc.) strongly affect the behavior of the system. Due to the lack of these information, some results shown in the manuscript are difficult to interpret.*

We have added the information requested by Reviewer in the text. The changes are described in detail in the answers to specific questions below.

*- How do the authors control or constrain the allowable corrections to the control variables? Since the spatial pattern of the correction to the 2 m air temperature (Fig. 8) is quite similar to the bias of the free run (Fig. 3), I have a concern that the system changes the control variables without reasonable constraint. The authors cite Köhl (2015) as a reference for the*

*atmospheric condition. In this paper, the errors of atmospheric field are prescribed by the standard deviation of the NCEP field. What does this mean? The standard deviation of the NCEP field is the ensemble spread of the NCEP climate model? If so, the standard deviation does not provide error of the reanalysis field, but provides the magnitude of the natural variability of the modeled climate system. Since the adjoint method tries to impose all the deficiencies of the model status to the control variables, the validity of the allowable corrections to the control variables should be carefully examined.*

The constraint on the control variables is separated into a mean and a time varying part. For the time varying part we use the standard deviation of the NCEP fields. Arguably, this choice would reflect a very pessimistic view of the NCEP data. The reason for this generous choice is that parameters are updated on a daily frequency and background information are mapped fields, which render the data of the constraints much larger than the actual ocean or sea ice data. Since errors of the controls are correlated in space and time, the actual degree of freedom is much smaller than the number of data provided for the constraints. The generous error in constrains compensates for the lack of correlation in the error weights. We believe that the standard deviation gives a very good approximation for the relative errors, although not for the absolute errors. The posterior evaluation of the corrections does not reveal any unrealistically large change in parameters which would point to a critical influence of the weights.

We added the following information in the text:

"For the atmospheric control variables, uncertainties were specified as the maximum of the STD of the NCEP fields and the errors for the mean components of air temperature, humidity, precipitation, downward shortwave radiation and wind were specified as 1°C, 0.001 kg/kg, $1.5 \times 10^{-8}$ mm/s, 20 W/m$^2$ and 2 m/s, respectively. For the downward shortwave radiation, both mean and time varying parts were set to 20 W/m$^2$."

*- What is the new technical achievement or new scientific finding(s) of this study, particularly, in comparison with the number of former sea ice (and partly ocean) data assimilation studies for Arctic region? Since the sea ice dynamics of the adjoint model is not consistent with the forward model and the assimilation window is only 1 year, I hardly find any advantages on this experiment compared to the former studies using another assimilation method (e.g., optimal interpolation, 3D-Var, Green function method and EnKF).*

We perform the first multi-year adjoint data assimilation for the coupled Arctic Ocean sea-ice system. To our knowledge, only one other Arctic data assimilation exercise exists that assimilates sea ice and hydrographic information based on the adjoint method (Fenty et al., 2015). But those authors first assimilate hydrographic data and then sea ice, while we do both simultaneously. The adjoint method, as explained in detail by Fenty et al. (2015) and also in our text, is able to adjust the model in a dynamically consistent way, which the other methods (except the Green's function) are not able to do. The Green's function method, due to its limited amount of adjustable parameters, is at the edge of being an ocean-sea ice synthesis method. Although it does assimilate data, the influence of the few parameters is so limited that they are mostly only able to improve the climate of the model. Being a pioneering work, our effort is certainly not without potential for further improvement; however providing a description of the experience and lessons we have learned in the process will certainly be important and useful for the data assimilation community, which will pick up on this work and carry it forward.

As we already mentioned, the Reviewer's criticism of the sea ice dynamics in the adjoint not being consistent with the forward model is due to the wrong interpretation of our (faulty) model description, in particular, the assumption that the sea ice is not moving in the adjoint. In reality the sea ice in the adjoint model is in the free drift mode. Simplification of the adjoint model is a standard technic and similar modifications in the adjoint model were made, for example, by Fenty et al. (2015).

In summary:
- we believe that in our experiment design we use practices that are common in the adjoint assimilation community and which lead to dynamically consistent state estimates of the ocean-sea ice system.
- for the first time pan-Arctic multiyear coupled ice-ocean adjoint state estimate is performed and results are of significant interest for the data assimilation community and, in particular, to researchers dealing with the adjoint method.

**4. Minor points**
*- Page 4, line 16: Which module is excluded from the adjoint code? What effect do the authors expect by this exclusion, particularly, in relation to the conclusion of this study? –*

We modify the paragraph to explicitly mention the code modules inclusion/exclusion and possible effects:

"The adjoint model was modified here similarly to Köhl and Stammer (2008) to exclude KPP modules and increase diffusivity values compared to the forward run. This is done to avoid exponentially growing adjoint variables. The sea ice module was active in the adjoint integration, but the part of the sea ice dynamics which treats rheology was switched off, so that the sea ice model was in a free drift configuration. This approach led to a reduced (approximate) adjoint producing smoother adjoint gradients. These gradients can still be successfully used to improve the large scale state of the model (see Köhl and Willebrand (2002) and Köhl and Stammer (2008) for more details). Similar simplifications of the adjoint model were used by Fenty at al. (2015) and Liu et al. (2012) provided an evaluation of the effect of modifications in the parameterizations on the adjoint. They confirm mostly small changes, although regionally some patterns of the gradients may shift. Since the gradients are only a means to find the cost function minimum and the forward code (and thus the minimum itself) is unmodified, changes to the gradient may lead to lower performance in finding the minimum but not to different states once the minimum is found."

*Page 4, line 16-17: Why the sea ice dynamics are switched off? Due to the absence of sea ice dynamics, the adjoint model cannot provide the optimal gradient, and therefore the correction to the control variable may not reflect (modeled) reality.*

As mentioned in the response to the major points, we have switched off only the rheology of the sea ice model, so that the sea ice in the adjoint model was in a free drift mode. We apologise for the confusion this might have caused since it might have given the impression that the adjoint variables to sea ice don't move at all.

As explained already in the answer to the previous point, we believe that simplifying the adjoint of the forward model is a necessary condition to get gradients that are instrumental in effectively reducing the cost function. We are not aware of any realistic long-term ocean adjoint data assimilation study that uses the full adjoint of the forward model.

*- Page 4, line 20-23: Are the control variables used to define penalty term in the cost (object) function? Please provide the exact definition of the penalty term for reproducibility of the study.*
*- Page 4: Please describe exact form of the cost function (including penalty terms) used in this study.*

We have added a brief definition of the cost function in the text, but we point the reader to Fenty et al. (2015) for more details, who have identical cost function formulation. The text now reads as follows:

The cost function J is defined as follows:

$$J = \sum_{t=1}^{t_f} [y(t) - E(t)x(t)]^T R(t)^{-1} [y(t) - E(t)x(t)] +$$

$$v^T P(0)^{-1} v + u_m^T Q_m^{-1} u_m + \sum_{t=0}^{t_f-1} u_a(t)^T Q_a(t)^{-1} u_a(t) \quad (1)$$

where $y(t)$ is a vector of assimilated data in time t, $x(t)$ is a vector of the model state, $E(t)$ is a matrix which maps the model state to the assimilated data, $v$ is a first guess initial condition, $u_m$ is a mean atmospheric state and $u_a(t)$ is a time-varying atmospheric state. Additional weights $R(t)^{-1}$, $P(0)^{-1}$, $Q_m^{-1}$ and $Q_a(t)^{-1}$ control the relative contribution of different terms in the cost function. More detailed description of the cost function and optimization procedure can be found in Fenty et al. (2015).

*- Page 4, line 23: How did the authors define the uncertainty for hydrographic (e.g., EN3, NISE, etc.) and satellite data? Although the author cited Köhl (2015) as a reference, the representation error of in-situ measurements should be different since the resolution of the model differs.*

The representation error unfortunately will remain the same for two reasons. First, in comparison to a Rossby radius of less than 5 km in the Arctic, a resolution of 16 km is still not sufficient to resolve eddies and the related processes in the Arctic. In relation to the Rossby radius, the Arctic configuration is probably of similar resolution as the global configuration on average. Second, even for a truly eddy resolving version, the problem of the representation error remains for all assimilation windows longer than a few eddy turnover times scales, because on these longer time scales we loose the ability to reconstruct individual eddy development and movement due to the chaotic dynamics. The eddy field becomes statistical and only an adjustment of the statistical properties remains feasible (see Köhl and Willebrand (2003) on how this can be achieved).

*- Page 4, line 24: Why did the authors apply constant error of 50% for sea ice concentration? OSI-SAF provides uncertainty estimates of ice concentration at every grid points. The constant error does not take into account the large uncertainties over the marginal ice zone, while it underestimates weight of reliable data over the central arctic.*

We added the following explanation in the text:

"We verified the sensitivity of our results by using space-time varying sea ice uncertainty estimates as they became available, as well as different values of a constant error. Results of

the sea ice assimilation with variable uncertainties were very similar to the ones with a constant error value of 50%."

*- Page 4, line 24: Why the data assimilation is performed in one year chunk? Due to this experiment design, the authors cannot take the advantage of adjoint method. Particularly, the authors cannot examine the effect of data assimilation to the ocean variables, since 1 year chunk is too short even for layers above Arctic halocline.*

The use of one year chunks indeed has limited our ability to get improvements related to the long-term ocean variability. However, we disagree with the Reviewer's strong statements that "*the authors cannot take the advantage of the adjoint method*" and "*cannot examine the effect of data assimilation to the ocean variables*". We believe that our study actually demonstrates the opposite. A one year time scale seems to be enough to successfully assimilate sea ice concentrations, which, as mentioned in the subsection "2.2 Adjoint data assimilation approach", is the main focus of the study. It is also enough to considerably alter the surface layers of the ocean, which are most important for the short-term ocean-atmosphere exchange.

We added the following text to the manuscript:

"The use of one year segments is related to technical reasons; we are not able to get useful sensitivities for a time period longer than a year for all years of our 2000-2008 assimilation period. We were successful in completing a 2-year assimilation at one occasion (2005-2004), but the results for sea ice area and thickness were not noticeably different from the 1-year chunk assimilation."

Getting stable and useful adjoint gradients on longer time scales for sea ice concentration is a challenge, which to our knowledge groups around the world did not solve to date. Sea ice is a faster moving medium compared to the ocean and in addition is not a smooth global field, making it hard to handle in the adjoint. This study exploits achievements in adjoint sea ice assimilation that are currently available. We believe that in our manuscript we have demonstrated that, even for short assimilation periods, the use of the adjoint method in the Arctic is useful.

*- Page 5, line 8-10: How did the authors define the relative weight between different types of observation? Satellite measurements generally cover the large area with constant time interval, whereas the errors of the data are not independent but covariant. On the other hand, in-situ ocean measurements are very sparse, while the measurement errors are almost negligible (and therefore assumed to be independent each other) compared to the representation error. In addition, the magnitude of the representation error depends on the size of model's grid cell. How did the authors handle these issues? Description is needed.*

Little is known about the covariance of errors, and error covariances for data terms are difficult to implement into the adjoint method (a feature not implemented so far). For most of the data, the error covariance is not a large problem because the data is sparse. Exceptions are mapped data that are processed via objective analysis. We took care of the reduced degree of freedom for the climatological data by increasing their error by a factor of two. All other data errors are not adjusted since only relative errors matter for the cost function and similar constraints apply for all data. For instance, in situ data error is correlated with depth while along track data is correlated along the track. Lacking the ability to specify anything clearly better, we settled with the simplest approach to assume that reduction in degrees of freedom is similar across all data types.

*- Page 6, line 23-25: I don't understand the meaning of this sentence. More explanations are needed.*

We rewrote the sentence as follows:

"Both metrics suffer from the inability to guarantee that improvements in this metric also lead to an overall improved match in the spatial sea ice coverage, since a perfect total SIC or SIE evolution may still correspond to considerable differences to the data in their regional distribution."

*- Page 6, line 28-30: Please explain relation between the Hausdorff distance (Dukohvskoy et al. 2015) and the metrics used in this study (sum of the RMS errors). I do not understand why the authors introduced Dukohvskoy et al., (2015) here.*

The reviewer is correct – we mentioned the Hausdorff distance from Dukohvskoy et al. (2015) without actually using it. We therefore removed the two last sentences in the paragraph and moved the reference to Dukohvskoy et al. (2015) to the end of the previous sentence:

"This calls for changing the common practice of model evaluation by only comparing their ability to simulate present day SIE without considering the sea ice spatial distribution (e.g. Dukhovskoy et al., 2015)."

*- Page 7, line 18-24: It is hard to believe that the spatial pattern of the bias of 2 m air temperature of NCEP reanalysis coincide with that of modeled ice concentration bias. If I understand correctly, the uncertainties of the NCEP reanalysis data used in this study are given by the standard deviation of ensemble runs of NCEP climate model. Does the difference between the ensembles has such a sharp gradient like the correction to the 2 m air temperature (Fig. 8 the first row, left)?*

The NCEP reanalysis uncertainties were determined as the standard deviation of the whole NCEP time series not as deviation of ensemble runs of the NCEP model. We are not aware of any ensemble runs of the NCEP RA1. This reanalysis is based on 3DVAR and does not produce an ensemble as part of the method. Maybe the reviewer has a different reanalysis in mind, but why would results from that be more appropriate? In any case, the shape of the correction field should not correspond to the uncertainties field of the control variable because corrections are related to the errors, while uncertainties describe only statistical properties of errors. Values of the corrections of course should be within the range of the uncertainties, which is the case.

*- Page 7-8: The authors described that the correction to the 2 m air temperature is the main driver to improve the sea ice concentration in the assimilated field, and the contributions from other control variables, such as wind forcing, are very small. I am afraid this may be an artifact due to the lack of sea ice dynamics in the adjoint model, as described in major point.*

As described in the answer provided above to the major points, our previously incomplete description must have led to the assumption that the adjoint variables to the sea ice are not moving at all, while the actual approximation in the adjoint would lead to the opposite effect. We completely understand the Reviewers' confusion and therefore we tried to make description of the adjoint formulation clearer. We have also considerably modified the section

"Control variables" and the statement about the relative contribution of thermo-dynamical forcing to the improvement of the model state is removed.

*- Page 7, line 31-33: The wind can play a role not only in local redistribution of the sea ice along the shore and ice edge, but also in large-scale sea ice distributions, although such effect is not seeable in the present experiment design.*

For the case of the free drift used in our work we expect that the gradients come from a model in which the sea ice is actually more responsive to the wind forcing compared to the case where the rheology was switched on. In the new version of the "Control variables" section, the statement the Reviewer is referring to was removed.

*- Page 8, line 8: ".., both making atmospheric forcing actually worse". I do not understand the meaning of this sentence.*

We modified the text in the following way:

"But it could equally also point to problems of the correct attribution of sea ice concentrations from satellite data. In both cases, corrections to atmospheric control variables will not improve the quality of the original atmospheric forcing, but on the contrary may make it worse."

*- Page 9, line 24-25: This is interesting. Could the authors provide specifications of the mechanism?*

In our opinion the most probable reason for the slight reduction of the positive temperature bias in the Eurasian Basin of the Arctic Ocean is a modification of the Atlantic Water upstream, before it enters the Arctic Ocean. Polyakov et al. (2005) estimated the travel time of the Atlantic Water temperature anomalies between the Svinoy section and Fram Strait to be about 1.5 years. Taking into account the relatively good observational coverage of the North Atlantic Ocean, even with short assimilation periods, the modifications of the near-surface ocean layers before they enter the Arctic Ocean and dive under the halocline can be enough to alter properties in the deeper layers of the Eurasian Basin. In other words, the model probably fixes the problem of too warm Atlantic Water entering the Arctic, and the reduced temperature bias in the Arctic Ocean itself is a consequence.

*- Page 9, line 33 – Page 10, line 1: How much is the ratio of relative contributions to the cost function between ocean variables and sea ice variable? If the contribution from sea ice variables dominates, the system tries to change the control variables which have large impact on sea ice, and then it is natural that the changes of the ocean variables are small.*

In Section 2.2 we wrote the following:

"Taking into account differences in the amount of sea ice concentration and sea surface temperature data compared to the amount of hydrography data, it is not surprising that most of the contributions to the total reduction of the cost function are from SIC and SST. Hence, most of the improvements can be expected to happen in these fields, while changes in the state of the ocean are expected to be small."

However, the reviewer is correct in that a similar statement is appropriate in the discussion about ocean changes. We therefore modify the text as follows:

"This is probably due to the fact that the volume flux is mostly controlled by the wind stress, which means that the corrections of the control variables discussed above do not contribute considerably to changes in the ocean circulation. This is expected since the amount of sea ice concentration data is much larger than the number of hydrographic observations in the Arctic Ocean, so that the assimilation system tries to change control variables in a way that will have larger impact on the sea ice. However, episodically, significant changes can be observed (for example in summer 2008) when modifications in the throughflows at Fram Strait are noticed, which are about 60% larger than in the forward simulation (Fig. 11a)."

*- Page 10, line 1-3: The fluxes shown in Fig. 11 are improved by the assimilation? As far as I know there are some flux estimates through Arctic gateways based on observations (e.g., Tsubouchi et al., 2014, JGR).*

Thank you for pointing us to this publication. However, the flux estimates presented in Tsubouchi et al. (2012) are for a very short period of time (August-September). We therefore calculated fluxes for the same period of time and added them to the table, along with estimates from Tsubouchi et al. (2012).

The following text was now added to the manuscript:

"We also show mean fluxes for August-September of year 2005 and compare them to the results of Tsubouchi et al. (2012), who applied an inverse model to data obtained in summer 2005 to calculate net fluxes of volume, heat and freshwater around the Arctic Ocean boundary."

"Considering Tsubouchi et al. (2012) to be a good approximation of observed values in August-September 2005, it is hard to definitely conclude if ocean fluxes become better or worse after the assimilation (Table 2). Some values, such as the volume flux through Davis Strait and the Barents Sea Opening, or the freshwater flux in the Fram and Davis Straits, have changed and became closer to the values of Tsubouchi et al. (2012). Other values moved even further away from their estimates."

*- Page 10, line 25-28: see the comments above.*

We removed the statement about the contribution of 2m temperature to the improvement of the sea ice state.

*- Page 11, line 3-4: This result is interesting, but I am still afraid that this improvement might be achieved by wrong reason, due to the absence of sea ice dynamics in the adjoint, since the ice dynamics is important for the redistribution and accumulation of sea ice.*

This comment again is related to the erroneous interpretation of our model setup description (see several comments above) by assuming that the sea ice in the adjoint model is not allowed to move.

*- Page 11, line 11-13: I agree that the estimated state of sea ice is consistent with the modeled physics, whereas due to the lack of ice dynamics in the adjoint, we cannot make sure the correction to the control variables are realistic. In other words, we cannot exclude a*

*possibility that the estimated state is achieved by artificial forcing different from reality, and therefore by thermodynamic and dynamical balance different from reality.*

Since the forward model is uncompromised by the approximations made in the adjoint, the effect of these approximations is always secondary, i.e., that a minimum cost function has not been found. In any case, the forward model is certainly also flawed in many ways and there is no guarantee that the estimated state is not achieved by an artificial forcing that has little to do with reality. Our comment at the end of Section 4 is exactly about that. For instance, there are many processes, like tides or ice-wave interaction that are not included in the forward model, which may be responsible for certain biases of the model and ultimately in the estimated atmospheric state. We, of course, act in the framework of approximations included in our model configuration and can only make conclusions about this system.

*- Figure 3 caption, line 3: "third" should be "forth".*

Thank you. This was fixed.

*- Figure 3, 4, 8 and 10: It would be helpful for comparison of the spatial patterns, if the longitude and latitude lines (as in Fig. 9) are embedded in these figures.*

To comply with the Reviewer's request, we redraw Figures 3, 4, 8 and 10 to add longitude/latitude grid lines.

---

## Author Comment (AC2) · 5 Jun 2017

**Reviewer 2**

**Summary**

*In this work the authors demonstrate the synthesis of hydrographic and sea ice concentration data into a 16-km horizontal resolution Arctic and North Atlantic coupled sea ice-ocean model. The reduction of an uncertainty-weighted model-data difference cost function was achieved by iteratively optimizing a set of adjustments to a set of atmospheric and initial condition control variables using gradient information provided by the adjoint of the numerical sea ice-ocean model. The final multiyear state estimate was constructed by optimizing each single year between 2000 and 2008 in succession - the final optimized state of year X is defines the initial state for year X+1. The authors demonstrate improvements of the model's reproduction of the data. The largest reduction in terms of percentage is found with sea ice concentration and SST with lower relative cost reduction for other data, including T and S profiles, SSH, and mean dynamic topography. The largest sea ice concentration cost reductions in terms of RMS are found during summer months. Discrepancies between simulated and observed sea ice extent are found to increase in some months even when discrepancies in simulated and observed total sea ice area decrease. After synthesizing ocean and sea ice data, little impact is seen in ocean volume, heat, and freshwater fluxes through Fram Strait and Davis Strait.*

We thank the Reviewer for the thoughtful evaluation of our manuscript. In the following Reviewer's comments are in italic, our answers are in usual font, text from the manuscript is in the quotation marks and the new text is in blue color.

**Specific Comments**

*1) With respect to the title, assimilation is not "into a Coupled Ocean-Sea Ice Adjoint Model". The assimilation is "into a Coupled Ocean-Sea Ice Model using its adjoint".*

We thank the Reviewer for this suggestion and changed the title accordingly.

*2) Abstract: Better to provide the actual spatial resolution of the satellite sea ice concentration data that is assimilated rather than refer to it as 'high resolution'.*

We decided to remove the reference to the resolution altogether because the sea ice data are assimilated, as the other data, on the model grid. The sentence now reads as follows:

"Satellite sea ice concentrations (SIC), together with several ocean parameters, are assimilated into a regional Arctic coupled ocean-sea ice model covering the period 2000-2008 using the adjoint method."

*3) Page 1, Line 6: 'values of sea ice extent become underestimated' doesn't define a metric. Is the metric the sum of model minus data or weighted model minus data difference or the RMS of model minus data or something else?*

We have tried to make the statement more precise, now it reads as follows:

"During summer months, values of sea ice extent (SIE) integrated over the model domain become underestimated compared to observations,…"

*4) Page 1, Line 6-7: Characterizing a state estimate of a system as complex as the Arctic Ocean requires that one analyzes a suite of metrics. The author's statement that one the sea ice extent metric is "not suitable to characterize the quality of the sea ice simulation" is odd and out of place. To whom is this statement aimed? This seems to be a straw man argument.*

The statement is more related to the practice of characterising the quality of a sea ice simulation (not the complete Arctic system) in the model by only considering one metric, namely the integrated Northern Hemisphere September sea ice extent. This is still common in many publications and authors are also guilty of this sin. However, we agree with the Reviewer in that the abstract is not the right place for such a statement and now the sentence reads as follows:

"During summer months, values of sea ice extent (SIE) integrated over the model domain become underestimated compared to observations, however the root-mean-square difference of mean SIE to the data is reduced in nearly all months and years."

*5) Page 1, Lines 10-11: The atmospheric control variable adjustments that one finds during any optimization are intimately related to the magnitudes of the prior uncertainties of the individual terms of the first-guess atmospheric state. The author's statement that biases in sea ice are reduced 'mainly due to corrections to the surface atmosphere temperature' is difficult to interpret because the reader does not know the magnitude of prior uncertainties used during e optimization. Are you referring to the sum of the squared normalized adjustments? How is surface atmosphere temperature identified as the main control variable correction since atmospheric forcing units are arbitrary?*

As suggested by the Reviewer, we have re-evaluated our analysis of the corrections to control variables and decided to remove the statement about the 2-m air temperature contribution to the improvement of the model state.

*6) Page 2, Line 8. The authors may consider using the term 'state estimation' to describe the model-data synthesis methodology used in this study instead of the term 'data assimilation'. An uninformed reader may think that the work conducted here referring to sequential data assimilation, a technique that has been applied to sea ice data for decades. The adjoint method used in this work is rather special and yields quite a different product (namely a physically-consistent ocean and sea ice state). Below I post an excerpt from Wunsch and Heimbach, 2007 in which they argue for their choice of the term 'state estimation' when describing the application of the adjoint method to combine data with a model (emphasis mine): "In physical oceanography, the problem of combining observations with numerical models differs in a number of significant ways from its practice in the atmospheric sciences. It is these differences that lead us to use the terminology "state estimation" to distinguish the oceanographers' problems and methods from those employed under the label "data assimilation" in numerical weather prediction. "Data assimilation" is an apt term, and were it not for its prior use in the meteorological forecast community, it would be the terminology of choice. But meteorologists, faced with the goal of daily weather forecasting, have developed sophisticated techniques directed at their own particular problems, along with an opaque terminology not easily penetrable by outsiders. Because much of oceanography has goals distinct from forecasting, the direct application of meteorological methods is often not appropriate."*

The term "adjoint data assimilation" is used in the community and even the usage of the term "state estimation" exist for applications with the Kalman filter. Since there is no agreement in the community, the reader has to anyway carefully read the methods section of the paper in order to understand how exactly data were used to improve the model. We agree with the Reviewer that the term "state estimation" is probably better to put the reader on the right path. However, "state estimation" inherited the flavour of trying to estimate a static climatological state, as it was attempted in the first applications of the adjoint method during the WOCE era, therefore we find it to be a less appropriate term; but again, current usage of both terms show that the nomenclature is not well defined and although we could live with the term "state estimation", we don't find it really better.

*7) Page 4 Line 10-11: List the control variables.*

The list of all control variables can be found in the text below. We did not use the longwave radiation as a control variable for final simulations. We adjusted the text accordingly.

*8) P4 Line 20: Describe why the atmospheric control variable frequency was changed to daily.*

We incorrectly used information from a different experiment setup, so the frequency of updates is actually once per three days, which is still higher than 10 days used by Köhl (2015), who has chosen 10 days due to computational (memory) limitations. We added the requested information and now the sentence reads as follows:

"In contrast to Köhl (2015), additional control variables are optimized and the frequency of the updates is enhanced to once per 3 days in order to reflect shorter time scales of sea ice variability."

*9) P4 Line 23: As atmospheric adjustments are an important control parameter in this work, the authors should (a) explicitly state how they were derived as Kohl (2015): "For the atmospheric state, errors are calculated as before from the [standard deviation] of the NCEP fields." And (b) show maps of their magnitudes in the main text or in supplemental materials. Also, because they are so important, more discussion about your choice of standard deviation of NCEP fields is appropriate. The standard deviation of Arctic near-surface atmosphere temperatures is considerable given the large seasonal cycle. In much earlier versions of ECCO/GECCO the use of atmospheric state standard deviations could be justified because in mid-latitudes and the tropics they partially captured "random" variations due to synoptic variability. At high latitudes the standard deviation for near-surface atmosphere temperature and shortwave radiation is mostly due to the seasonal cycle.*

Although there is a large seasonal cycle the difference between the STD with and without the seasonal cycle is actually not that large for most of the globe; but it is true that in the Arctic region the error is with values around 12-30°C overestimated by a factor of 2. The STD is in both cases relatively homogeneous, such that a figure would not provide valuable information.

We have added the following text to the manuscript:

"For the atmospheric control variables, uncertainties are specified as the maximum of the STD of the NCEP fields and the errors for the mean components of air temperature, humidity, precipitation, downward shortwave radiation and wind were specified as 1°C, 0.001 kg/kg,

$1.5 \times 10^{-8}$ mm/s, 20 W/m$^2$ and 2 m/s, respectively. For the downward shortwave radiation both mean and time varying parts were set to 20 W/m$^2$."

*10) Page 4, Line 24: Why are the sea ice data assigned a constant 50% error? Satellite SIC products have errors that are far smaller than that everywhere except in the MIZ and in summer when meltponds are present.*

We added an explanation in the text:

"We verified the sensitivity of our results by using space-time varying uncertainty estimates as they became available, as well as different values of a constant error. Results of the sea ice assimilation with variable uncertainties were very similar to the ones with constant error value of 50%."

*11) Page 4, Line 27: To clarify, each year after the first uses initial conditions that are identical to the final state of the previous year, correct?*

Yes, this is correct. We feel that the sentence: "After the first year assimilation, we move to the next year using the final state of the previous year's successful iteration as initial conditions.", describes this sufficiently.

*12) Page 4, Lines 3-4: Some SST products have nonzero values beneath sea ice. Is that the case in the RSS dataset?*

The RSS data have a "sea ice" flag that, in practice, means missing value. We didn't take the SST data if the "sea ice" flag was set.

*13) Table 1: I understand that the PHC climatology had large biases relative to modern Arctic T and S because it was derived with observations mainly from the 1970's and 1980's and before the recent shifts in Arctic heat and freshwater (McPhee et al, 2009). Can you comment on how the simultaneous use of the PHC climatology alongside contemporary data may have affected the T and S cost reduction?*

We used the PHC climatology only for model initialisation and it was not used in the data assimilation. We have removed it from Table 1. Since the model was started from PHC and in situ data is sparse, the model cannot be corrected very much away from the first guess, a point made in the text.

As mentioned below in the paper, the changes of the deep ocean state are quite small due to both the predominance of sea ice and sea surface temperature observations over interior hydrographic observations and the short assimilation periods (yearly chunks). Using a more recent Arctic Ocean state as initial conditions would certainly be beneficial due to an initially smaller cost in T and S. However, judging from the experience we gained during this exercise, we believe that there would still be hardly any significant cost reduction of T and S beyond surface layers in the Arctic Ocean, mainly again because of our experiment design and a much larger amount of sea ice data compared to hydrography data.

*14) Page 4, Paragraph 1: Cost function reduction percentages are important but obviously they are dependent on how close to the data you were when you began your simulations. The first-guess solution of Fenty et al., (2015) could have been further from the data than your*

*first-guess solution. While both may end up in the same state, their reduction percentage would be higher. The most important information is how well one's final state estimate fits the data. Much less important is the magnitude of the improvement relative to one's (somewhat arbitrary) starting point.*

We agree that just stating the percentage reduction is problematic, but it is nevertheless important information about the performance of the assimilation. We cannot really assume that we found a minimum, and therefore success is usually evaluated by the amount of reduction. It would not be a too bad assumption that both controls perform about equally well. We have added the caveat of this comparison in the text:

"In 2004 the cost reduction of sea ice area was about 30%, less than that reported by Fenty (2015) (49%), which may partly be explained by differences in the first guess solution."

*15) Page 4, before line 31: It may be useful to mention how many iterations were conducted before the 1% threshold was achieved. In Figure 3 I see "iteration 3" as the final iteration for 2005 and 2007. That strikes me as unusual. If your cost was dominated by SIC and SST data and the adjoint method quickly reduced the misfits of those data, then I can see how you hit the 1% total cost reduction threshold quickly. However, it is possible that if those two datasets were ignored, the adjoint machinery could have continued to substantially reduce misfits in other datasets. Can you comment on that?*

The Reviewer is correct. The cost is dominated by SIC and SST. These data easily respond to the surface controls. We added this explanation:

"The cost is dominated by SIC and SST data, which easily respond to the surface controls, and the adjoint method quickly reduced the misfits of those data, so that the number of iterations was usually less than five."

*16) Page 5, Line 15: There may be a missing figure. I cannot match up Figure 2 to the description offered here. Fig 2 is % cost reduction in different years vs. data.*

Thank you very much for spotting this. The first paragraph of the section "Sea ice concentration changes" was from a previous draft version. We did not intend to include it in the manuscript. It is now removed.

*17) Page 5, Line 24: Good to additionally mention why most models overestimate sea ice in the Greenland Sea with a reference.*

This statement was referring more to results of climate models (see for example Figure 9.23 in IPCC AR5 Chapter 9). It is not correct to transfer this result to regional ocean-sea ice models because much of the bias in climate models result from biases in the atmosphere. So we have removed this part of the sentence. Now the end of the sentence reads as follows:

"Most noticeable is the decrease in the SIC along the east coast of Greenland after data assimilation"

*18) Page 5, Line 35: This is probably because in these extreme months the location of the sea ice edge is relatively stable compared to spring and fall months when the ice pack contracting and expanding.*

We thank reviewer for this explanation, which we added in the text with a slight modification. Now the text reads as follows:

"Interesting to note, values of RMSE in March and September are quite similar, despite the large differences in ice cover in the two months. One of the possible reasons is that the location of the ice edge in those extreme months is relatively stable compared to spring and fall when the ice pack is contracting and expanding."

*19) Section 4: This entire discussion must be rewritten. Atmospheric control variable adjustments seem to be compared by their relative magnitudes but their relative magnitudes are not meaningful because these physical variables have different, arbitrary, units. By all means show the magnitude of the adjustments but to make a meaningful comparison one should first normalize them by their prior uncertainties. a. This includes Figure 8, which should be updated to show all control variable adjustments normalized by their uncertainties. Also include longwave radiation.*

Neither dimensional, nor normalized values provide the impact of the changes per se. The prior errors of controls have nothing to do with the impact or even the anticipated impact but describe only our knowledge about them. Moreover, our choice of STD does not make a difference because there is no reason why one STD of perturbation should have a similar impact across all parameters.

Nevertheless, in the optimization the parameters enter normalized, and corrections are generated according to the normalized sensitivities and the approximation of the Hessian matrix. Since we have only a few iterations completed, the Hessian stays not very far away from its initial value, which is the identity. Therefore, in this special case the normalized corrections will still more or less reflect normalized sensitivities. Since the impact is the product of the sensitivity and the corrections, normalized corrections will provide a reasonable measure of the relative importance of the parameters.

As suggested by the Reviewer we have added the normalized corrections to Fig. 8 and reworked the section. The additional text reads as follows:

"Dimensional values of the corrections do not directly provide information about the relative importance of changes in the controls for bringing the model into consistency with observations. However, due to the relatively small number of iterations, we can use values of the corrections normalized by uncertainties as a reasonable measure of the relative importance of changes in control parameters. Spatial distributions and monthly means of absolute values of normalized corrections for the year 2005 are shown in Fig. \ref{fig:8}.

Wind corrections seem to play integrally a larger role, with a maximum in May. This agrees well with results of \citep{Kauker2009}, who used an adjoint sensitivity analysis to determine the relative contribution of different atmospheric and ocean fields to the September 2007 sea ice minimum and found that the May-June wind conditions are one of the main factors in setting up extremely low sea ice conditions in Summer 2007. The maximum contribution of air temperature corrections occurs in June and it is about a factor of five smaller than the contribution of the wind corrections. However, using free drift in the adjoint biases the sensitivities towards larger sensitivities of sea ice to wind changes. Since measuring the impact by the normalized corrections relies on the assumption of correct sensitivities, the results may be also biased to too large an impact by the wind.

Given the absence of proper sea ice dynamics in the adjoint model (only free drift is used) and lack of many important processes in the forward model (such as tides or waves), the question remains to what extent corrections to control variables reflect deficiencies in the forcing fields or a compensation to the sea ice model or sea ice data deficiencies, particularly since in the Arctic the NCEP reanalysis seems to perform well near the surface \citep{Jakobson2012}."

Showing only normalized adjustments maybe tells a lot to people involved in the adjoint community, but for most people it's just easier to look at absolute values that have some physical meaning.

The long wave radiation was not a control variable in our final simulations, so we can't show it. We have changed the text in the method description accordingly.

*20) Page 8, Line 20-22. The "probably realistic" spatial distribution of the Kwok Arctic sea ice thickness field deserves a reference. Are the 0.7 m errors spatially correlated or uncorrelated?*

The 0.7 m is a mean error; the individual values would vary of course, depending on the sea ice thickness. Reference to Kwok et al. (2008) was added.

*21) Page 9, Line 22-24: Neither the length of the simulation nor the number of T/S profiles is a fundamental impediment to magnitude of model-data misfit reduction. An iteration 0 state with T and S close to the data as measured by the prior uncertainty could be responsible. Maybe averaged normalized costs should be added to Figure 2 for each cost category for iteration 0 and the final iteration.*

The reason why we think the number of data and time period of assimilation matter is that the data information has to be able to reach the sensitivities to the controls in the adjoint model. All data from year 2 and later is excluded from modifying the initial condition due to the separation into 1 year windows. The time window of one year, on the other hand, is too short for deep data signals to be able to reach the surface. Sparse data is in general a problem because, due to the lack of covariance information, sparse data is likely to produce unphysically small-scale corrections, which are likely to be not beneficial for the simulation of the dense SST and SIC data that determine most of the cost.

*22) I may be incorrect but it seems that no Arctic Ocean T and S pro- files were used in this work. I do not see Arctic Ocean data in the Ingleby and Huddleston report and the NISE database doesn't show data north of the Norwegian Sea. Given that the assimilation period overlaps with the existence of ice-tethered profilers, why were ice-tethered profile data not included (http://www.whoi.edu/page.do?pid=20781)? As for the CTD data in the Arctic, both the ICES database (http://www.ices.dk/marine-data/data-portals/Pages/ocean.aspx) and the World Ocean Database v3 (https://www.nodc.noaa.gov/OC5/WOD13/ ) have data for the time period considered in this work. There may be perfectly fine reasons for excluding these data but the reasons should be offered.*

At the time we have started our assimilation efforts (year 2012), the combination of the EN3 (which includes a good amount of Arctic Ocean T and S profiles) and NISE dataset was the best available option in terms of data coverage and technical efforts were required to interpolate observations to the model grid. Later we decide to stick with this choice for consistency.

We now added the following:

"The collection of hydrographic observational data in the Arctic Ocean used in the present work is not comprehensive and does not include, for example, ice-tethered profile data. In the present pilot study we decided to stick to two well-structured data sets available at the time we have started our efforts."

**Technical Corrections**
*1. Page 1, Line 5: change 'become' to 'are' as in 'values of sea ice extent are underestimated'*

Corrected.

*2. Page 1, Line 5: first comma to semicolon. Or split this long sentence into two before 'however'*

We changed it to semicolon.

*3. Page 1, Line 14: strike 'to date'*

Corrected.

*4. Page 1, Line 16: reference?*

We now cite Overland and Wang (2013).

*5. Page 1, Line 17: strike comma before 'is therefore of utmost importance'*

Corrected.

*6. Page 1, Line 24: strike 'if not possible'*

Corrected.

*7. Page 2, line 2, strike comma before 'the community'. Strike 'heavily'.*

Corrected.

*8. Your doi for Detlef's 2016 paper is incorrect. It should be DOI: 10.1146/annurev-marine-122414-034113*

We double checked and could not find a difference between the DOI that you have provided and what appears in the paper. Can you please specify what exactly is wrong in the DOI?

*9. Page 2, Line 19: strike "usually in general"*

Corrected.

*10. Page 5, Line 12: strike "are going to"*

Corrected.

*11. Page 5, Line 28: replace "very good" with "improved"*

Corrected.

*12. Page 5, Line 29: strike "thus"*

Corrected.

*13. Page 5, Line 31-32: This sentence deserves a rewrite for clarity. As mentioned above, relative percentage sea ice cost reductions are also a function of the (unknown) first guess states.*

The year in Fenty et al. (2015) is different as well as the first guess, so we remove the sentence completely.

*14. Add 'bears' before 'a good resemblance'*

Corrected.

*15. Page 5, Line 24: For clarity consider saying 'since a perfect total sea ice area evolution...' and the following sentence is redundant.*

We removed the redundant sentence and modified the sentence in question to comply with Reviewer 1 request as follows:

"Both metrics suffer from the inability to guarantee that improvements in this metric also lead to an overall improved match in the spatial sea ice coverage, since a perfect total SIC or SIE evolution may still correspond to considerable differences to the data in their regional distribution."

*16. Page 8, Line 20-22. Strike "except for the" and simply say that "Sea ice thickness are not provided by Kwok for the Barents and Kara Seas and the Canadian Archipelago because ..." with a reference.*

We deleted part of the sentence after "except for the" since it does not make sense to discuss uncertainty or realism of the data in the regions where they are not present.

*17. Page 8, Line 26: change "variables" to "variables'"*

We guess the Reviewer meant "to variable's". Corrected.

*18. Page 9, Lines 1-2: Why is it hard to provide quantitative estimates? You could plot time series of the uncertainty-weighted squared model-data misfit (normalized cost) before and after the assimilation.*

The Reviewer is right. Quantitative metrics are not hard to provide in general; we think the visual comparison of spatial distribution is more instructive than just a few numbers. We removed the respective sentence.

*19. Plotting model minus data or model minus data squared in Fig 5 might simplify comparison.*

Although your suggestion allows for an easier evaluation of the improvement, we decided to continue showing absolute values since we believe it is easier for most readers to interpret. Adding separate panels with differences would just duplicate the information and make the figure unnecessarily verbose.

*20. Section 4: Fonts on the time series of Fig 8 are also small and difficult to read. One subplot is cut off. After normalizing the summed control variable adjustments they could all be shown in the together in the same plot.*

We now made the fonts of the time series in Fig. 8 larger.

---

## Referee Report (RR1)

**Minor comments:**

(1) Page 5, Line 14-17: Regarding the assigned value of sea ice concentration data. I don't really understand the argument used to justify your 50% sea ice concentration data error when you say that you 'verified the sensitivity' of your results by comparing the 'results of the sea ice assimilation' between constant 50% errors and spatially and temporally-varying errors (from OSI-SAF?). If sea ice concentration misfits were the dominant (or only) term in the cost function then you could probably even use much higher concentration errors and end up with similar reductions of the cost function. So that exercise wouldn't verify the correctness of the SIC prior error. *A key aspect of the sea ice concentration prior error that seems to be missing in this paper is that the prior errors assigned to the data are central for determining whether your final state estimate is consistent with the data*. The statistics of the distribution of the model-data misfits of the state estimate should be consistent with the prior data error statistics, if they aren't then one needs to offer explanations.

When you use a 50% SIC error you are essentially saying that you would accept a distribution of SIC model-data misfits with a standard deviation of 0.5. Now, you might have a good argument for why you think your particular model would not be able to do better than that, but I don't see any such argument. I think a 0.5 error is very large if you consider all nonzero SIC points because so much of the Arctic has SIC near 1 for so much of the year. In winter in the central Arctic, both model and data are going to be so close that the RMSE errors are going to be very low, probably much lower than 0.5. Also, as far as I can tell there is no comparison of the SIC prior errors and the model-data residual statistics before and after the optimization.

You should probably show a distribution of the SIC residuals before and after the optimization and compare the standard deviations of those residuals against each other and against the prior error. With the 0.5 value that you assumed, you may find that you have formally achieved consistency with the data at iteration 0, or you may find that you achieve it after your iterations, or you may find that you have not achieved it. The RMSE tables offered are not sufficient because according to the text they include 'every grid location', which would include points where SIC in both the model and the data are always both 0.

I suggest that for each day separately you include only those points where the model OR the data have nonzero sea ice. If, before the assimilation, the model-data residuals RMSE each day are < 0.5 then you are already formally within your data prior errors and there is no apparent need to do data assimilation. If SIC model-data residual RMSEs are higher than 0.5 at iteration 0 then you have to determine how close to 0.5 they get after the assimilation. That's the point of the SIC prior error that is missing here. The SIC prior error defines a *target* for the model-data residuals that the state estimate is trying to achieve.

(2) Page 5 line 22: The 1% criteria that you used to stop iterating is not indicative of model-data consistency, it's indicative of a slowdown of the cost function reduction. Since only a

few years were considered, please mention then number of iterations required for each year to get to the 1% threshold as that information might be useful for future researchers.

You should also probably show the goodness of fit of your estimated state before and after the data assimilation compare with the prior error, especially with respect to sea ice concentration since that is the focus of the paper.  See comment above.

**Technical Corrections**

1. Page 4, line 9: should be 'ice-tethered profiler' data.  Also include a reference to ITP data here.
2. Page 5, line 10: write out standard deviation instead of STD.   Also, the standard deviation of the NCEP fields over which time period?
3. Page 7, line 12.  You probably mean 'since a perfect total SIA or SIE' instead of 'SIC or SIE'.
4. Your doi for Detlef's 2016 paper is still incorrect.  It should be DOI: 10.1146/annurev-marine-122414-034113   Remove the ncbi.nlm.nih.gov/pubmed link.

---

## Referee Report (RR2)

Review on "Sea Ice Assimilation into a Coupled Ocean-Sea Ice Model Using its Adjoint", by Koldunov et al., submitted for publication in The Cryosphere journal.

Page 3, line 25: change "Paremeterization" to Parameterization".

Page 4, line 15:

You define $v$ as the first guess initial condition. The way the cost function is written, it would seem that you would like $v$ to become as small as possible to reduce the cost function. I think you should define $v$ as the difference between the first guess initial condition and the model state at the beginning of the assimilation window: $v = x_b - x(0)$, where $x_b$ is the first guess (the background) of the model state at the beginning of the assimilation window. It should also be mentioned that the model states at times $t > 0$ are not control variables but are dictated by the strong constraint $x(t) = M(x(0))$, where $M$ represents the forward model.

A similar comment applies to the term with $u_m$: $u_m$ is defined as the mean atmospheric state. I don't believe you want to make $u_m$ as small as possible to reduce the cost function. Instead, you should define $u_m$ as the difference between the first guess mean atmospheric state and an optimized version.

Again, for the last term in equation (1), same comment applies.

Page 8, line 13: I believe SAT refers to surface atmospheric temperature but it has not been defined to this point.

Page 9, line 10: Please define the sea ice thickness you are using. Is it the average or the effective ice thickness ? The average sea ice thickness is defined as the volume of ice divided by the sea ice area. The effective ice thickness is defined as the volume of ice divided by the grid cell area. These 2 quantities are related like this:

$$average\ ice\ thickness = \frac{effective\ ice\ thickness}{sea\ ice\ concentration}$$

In light of this, make sure the comparison between ICESat ice thickness and the model output is done appropriately.

Page 10, line 10: "In June, considerable temperature differences cover a much smaller area…". Do you mean smaller than in September ? Please clarify the sentence.

Page 11, line 4: I guess the "forward simulation" here means the model run before the data assimilation, as it is specified later on the same page on line 19. Is that correct ? If possible clarify this in the text.

---

## Author Response (AR2)

**Reviewer #2**

We thank the Reviewer for the second evaluation of our manuscript.

*(1) Page 5, Line 14-17: Regarding the assigned value of sea ice concentration data. I don't really understand the argument used to justify your 50% sea ice concentration data error when you say that you 'verified the sensitivity' of your results by comparing the 'results of the sea ice assimilation' between constant 50% errors and spatially and temporally-varying errors (from OSI-SAF?). If sea ice concentration misfits were the dominant (or only) term in the cost function then you could probably even use much higher concentration errors and end up with similar reductions of the cost function. So that exercise wouldn't verify the correctness of the SIC prior error. A key aspect of the sea ice concentration prior error that seems to be missing in this paper is that the prior errors assigned to the data are central for determining whether your final state estimate is consistent with the data. The statistics of the distribution of the model-data misfits of the state estimate should be consistent with the prior data error statistics, if they aren't then one needs to offer explanations.*

*When you use a 50% SIC error you are essentially saying that you would accept a distribution of SIC model-data misfits with a standard deviation of 0.5. Now, you might have a good argument for why you think your particular model would not be able to do better than that, but I don't see any such argument. I think a 0.5 error is very large if you consider all nonzero SIC points because so much of the Arctic has SIC near 1 for so much of the year. In winter in the central Arctic, both model and data are going to be so close that the RMSE errors are going to be very low, probably much lower than 0.5. Also, as far as I can tell there is no comparison of the SIC prior errors and the model-data residual statistics before and after the optimization.*

Our argument was not that the 50% error is correct, but that the size of the error is unimportant for the result, at least for the tested range of errors. This does not necessarily hold for much larger errors; at some point the error will matter because ocean data will become the dominant factor in the cost function.

There are two roles of the error: one is consistency, the other the dependency of the result on the error. Typically, the latter is less critical because results of variational methods are known to be relatively robust against changes in errors. This is what we have verified. For the discussion of the results, the best error estimates should be taken into account. It is not so relevant whether the prior would have been consistent with the data given a 50% error. Relevant is instead if the final RMS agrees with the available error estimate. But we agree that this is a valuable piece of information, not only for those that use the synthesis product but also for those that generate the data.

We add the following paragraph to the text:

"In order to test the consistency of the estimate with the observations and the uncertainties we compare the spatial distribution of monthly mean sea ice concentration absolute differences before and after data assimilation to the maps of spatial distribution of monthly mean total standard error in the ESA SICCI sea ice concentration product ([ESA SICCI , 2013]). The latter provide daily spatially varying estimates of sea ice concentration errors. The

absolute differences after assimilation correspond well to the total standard error spatially and by value with only few spots along the edges with very high absolute differences (not shown)."

We show an example of such a comparison for September 2005 in the figure below only in the response to Reviewers.

[Figure]

*Fig. 1 Absolute differences of monthly mean sea ice concentration between model and observations before (iteration 0) and after data assimilation (iteration 3). Also shown are mean total standard error of the ESA-SICCI Sea-Ice product (https://icdc.cen.uni-hamburg.de/1/projekte/esa-cci-sea-ice-ecv0.html).*

*You should probably show a distribution of the SIC residuals before and after the optimization and compare the standard deviations of those residuals against each other and against the prior error. With the 0.5 value that you assumed, you may find that you have formally achieved consistency with the data at iteration 0, or you may find that you achieve it after your iterations, or you may find that you have not achieved it. The RMSE tables offered are not sufficient because according to the text they include 'every grid location', which would include points where SIC in both the model and the data are always both 0.*

*I suggest that for each day separately you include only those points where the model OR the data have nonzero sea ice. If, before the assimilation, the model-data residuals RMSE each day are < 0.5 then you are already formally within your data prior errors and there is no apparent need to do data assimilation. If SIC model-data residual RMSEs are higher than 0.5 at iteration 0 then you have to determine how close to 0.5 they get after the assimilation. That's the point of the SIC prior error that is missing here. The SIC prior error defines a target for the model-data residuals that the state estimate is trying to achieve.*

The RMSE tables contain sums of absolute differences for every grid point and every month. Since it is a sum, there is no artificial decrease in the RMSE happening, as implied by the Reviewer, due to the fact that when both model and data are zero there is no value to participate in the sum. It turns out that the term RMSE used in this context is confusing (since actually there is no mean involved) and we have renamed RMSE to the "sum of absolute differences (SoAD)" throughout the text.

*(2) Page 5 line 22: The 1% criteria that you used to stop iterating is not indicative of model data consistency, it's indicative of a slowdown of the cost function reduction. Since only a few years were considered, please mention then number of iterations required for each year to get to the 1% threshold as that information might be useful for future researchers.*

We added the following text:
"(it is 3 iterations for 2000, 2003, 2004, 2005, 2006 and 2007, 4 iterations for 2002 and 2008 and 5 iterations for 2001)"

*You should also probably show the goodness of fit of your estimated state before and after the data assimilation compare with the prior error, especially with respect to sea ice concentration since that is the focus of the paper. See comment above.*

The answer to this point is provided in the comment above.

*Technical Corrections*
*1. Page 4, line 9: should be 'ice-tethered profiler' data. Also include a reference to ITP data here.*

The references to Toole et al. (2011) and Krishfield et al. (2008) were added.

*2. Page 5, line 10: write out standard deviation instead of STD. Also, the standard deviation of the NCEP fields over which time period?*

The text was changed to:
"… standard deviation of the NCEP fields for the 1948–2008 time period…"

*3. Page 7, line 12. You probably mean 'since a perfect total SIA or SIE' instead of 'SIC or SIE'.*

Yes, thank you for spotting this. Changed accordingly.

*4. Your doi for Detlef's 2016 paper is still incorrect. It should be DOI: 10.1146/annurevmarine-122414-034113 Remove the ncbi.nlm.nih.gov/pubmed link.*

We have removed the PubMed link and the DOI has been now corrected to DOI: 10.1146/annurev-marine-122414-034113.

**Reviewer #3**

We thank the Reviewer for evaluation of our manuscript.

*Page 3, line 25: change "Paremeterization" to Parameterization".*

Corrected.

*Page 4, line 15:*
*You define $v$ as the first guess initial condition. The way the cost function is written, it would seem that you would like $v$ to become as small as possible to reduce the cost function. I think you should define $v$ as the difference between the first guess initial condition and the model state at the beginning of the assimilation window: $v = x_b - x(0)$, where $x_b$ is the first guess (the background) of the model state at the beginning of the assimilation window. It should also be mentioned that the model states at times $t > 0$ are not control variables but are dictated by the strong constraint $x(t) = M(x(0))$, where $M$ represents the forward model.*

*A similar comment applies to the term with $u_m$: $u_m$ is defined as the mean atmospheric state. I don't believe you want to make $um$ as small as possible to reduce the cost function. Instead, you should define $u_m$ as the difference between the first guess mean atmospheric state and an optimized version.*

*Again, for the last term in equation (1), same comment applies.*

That is correct. The differences to the priors are entering the cost function. The text was changed accordingly:

"$v$ is the difference between the first guess initial condition and the model state at the beginning of the assimilation period (only for the first year), $u_m$ is the difference between the first guess time mean atmospheric state and the optimized mean atmospheric state, $u_a(t)$ is the difference between the first guess time-varying atmospheric state and the optimized time-varying atmospheric state."

*Page 8, line 13: I believe SAT refers to surface atmospheric temperature but it has not been defined to this point.*

We changed the sentence to:

"Positive surface atmospheric temperature (SAT) corrections…"

*Page 9, line 10: Please define the sea ice thickness you are using. Is it the average or the effective ice thickness? The average sea ice thickness is defined as the volume of ice divided by the sea ice area. The effective ice thickness is defined as the volume of ice divided by the grid cell area. These 2 quantities are related like this:*

$$average\ ice\ thickness = effective\ ice\ thickness/sea\ ice\ concentration$$

*In light of this, make sure the comparison between ICESat ice thickness and the model output is done appropriately.*

The thickness the ICESat group provides is the average thickness. The nominal model output is an effective thickness, but we have converted it to the average thickness, so that the comparison is consistent.

*Page 10, line 10: "In June, considerable temperature differences cover a much smaller area…". Do you mean smaller than in September? Please clarify the sentence.*

We modified this part of the sentence, now reading as follows:

"In June, considerable temperature differences cover a much smaller area compared to September"

*Page 11, line 4: I guess the "forward simulation" here means the model run before the data assimilation, as it is specified later on the same page on line 19. Is that correct? If possible clarify this in the text.*

There is a definition of "forward run" in an earlier paragraph, which we have renamed to "forward simulation" for consistency. We also added an additional explanation to the line the Reviewer is referring to, now the sentence reading:

[revised manuscript text omitted]